

# Controllability, not chaos, key criterion for ocean state estimation

Geoffrey Gebbie[1] and Tsung-Lin Hsieh[1,2,3]

[1]Department of Physical Oceanography, Woods Hole Oceanographic Institution, Woods Hole, MA, USA
[2]Summer Student Fellow, Woods Hole Oceanographic Institution, Woods Hole, MA, USA
[3]Princeton University, Princeton, NJ, USA

*Correspondence to:* Geoffrey Gebbie (ggebbie@whoi.edu)

**Abstract.**

The Lagrange multiplier method for combining observations and models (i.e., the adjoint method or "4D-VAR") has been avoided or approximated when the numerical model is highly nonlinear or chaotic. This approach has been adopted primarily due to difficulties in the initialization of low-dimensional chaotic models, where the search for optimal initial conditions by gradient descent algorithm is hampered by multiple local minima. Although initialization is an important task for numerical weather prediction, ocean state estimation usually demands an additional task – solution of the time-dependent surface boundary conditions that result from atmosphere-ocean interaction. Here, we apply the Lagrange multiplier method to an analogous boundary control problem, tracking the trajectory of the forced chaotic pendulum. Contrary to previous assertions, it is demonstrated that the Lagrange multiplier method can track multiple chaotic transitions through time, so long as the boundary conditions render the system controllable. Thus, the nonlinear timescale poses no limit to the time interval for successful Lagrange multiplier-based estimation. That the key criterion is controllability, not a pure measure of dynamical stability or chaos, illustrates the similarities between the Lagrange multiplier method and other state estimation methods. The results with the chaotic pendulum suggest that there is no fundamental obstacle to ocean state estimation with eddy-resolving, highly-nonlinear models, especially when using an improved first-guess trajectory.

## 1 Introduction

The most complicated, and probably most realistic, numerical models of the ocean circulation are eddy-resolving ocean general circulation models (e.g., Arbic et al., 2010; Maltrud et al., 2010; Griffies et al., 2015). Such models are a natural choice in ocean state estimation, the combination of models and observations to reconstruct our best estimate of what the ocean has actually done (e.g., Stammer et al., 2002a). Here, we restrict our focus to state estimation as the transient reconstruction of the ocean state over a previous finite-interval of time where observations are available, following the convention of Wunsch et al. (2009). In order to unambiguously diagnose physical mechanisms of interest, the ocean state must be dynamically consistent: a solution to the dynamical equations of motion without any unphysical terms. The Lagrange multiplier method (e.g., Thacker and Long, 1988; Wunsch, 2010), sometimes called the adjoint method (e.g., Hall et al., 1982; Tziperman and Thacker, 1989), "4D-VAR" (e.g., Courtier et al., 1994; Ferron and Marotzke, 2003), or variational data assimilation (e.g., LeDimet and Talagrand, 1986;





Bonekamp et al., 2001; Bennett, 2002), is a method that satisfies both of these criteria, unlike the Kalman filter (e.g., Fukumori and Malanotte-Rizzoli, 1995) or nudging techniques (e.g., Malanotte-Rizzoli and Tziperman, 1995).

For the Lagrange multiplier method to be successful in state-of-the-art ocean models, two major issues need to be addressed: (1) the high dimensionality of the forward model and estimation problem, and (2) the nonlinearity of ocean models at increas-
ingly fine resolution. Research conducted by the ECCO (Estimating the Circulation and Climate of the Ocean) Consortium (Stammer et al., 2002b, 2004) has demonstrated that (1), the dimensionality of many million state variables, is not a fundamental problem. One caveat is that the convergence of the optimization process may be slower than hoped, but this is primarily an issue of computational efficiency. Regarding nonlinearity (2), the adjoint model has the same stability characteristics as the forward model, as the eigenvalues of linearized state transition matrix are the same as the transpose of the matrix (Palmer, 1996).
Therefore, nonlinearity in the forward model may be accompanied by an unstable adjoint model and Lagrange multipliers that grow exponentially with time. When the Lagrange multiplier method is used to enforce a nonlinear constraint such as a chaotic model, the search for a solution becomes iterative and the Lagrange multipliers provide gradient information that is used to minimize an objective function that describes the model fit to observations (e.g., Marotzke et al., 1999). For a bounded objective function with growing gradients, multiple local minima are present that complicate the search for a global minimum (e.g.,
McShane, 1989). Even sophisticated gradient descent algorithms such as the variable-storage quasi-Newton method (Nocedal, 1980; Gilbert and Lemaréchal, 1989) can become stalled in a local minimum and are not guaranteed to fit the observations adequately. For example, Lea et al. (2000) used the Lorenz (1963) model to conclude that the "adjoint does not tend to useful sensitivity values," echoing previous concerns with simple, chaotic models (e.g., Gauthier, 1992; Miller et al., 1994a; Tanguay et al., 1995).

Due in part to the concerns raised about nonlinearity in simple models, the method of Lagrange multipliers has rarely been applied to realistic models over time windows longer than the eddy scale. For example, some studies restricted the time windows to be short enough that unstable modes would not grow too large (e.g., Schröter et al., 1993; Cong et al., 1998). The Southern Ocean State Estimate was produced with an approximate version of the method of Lagrange multipliers, where the Lagrange multipliers are calculated by an adjoint model with artificially large diffusivities that stabilize the model (Mazloff
et al., 2010). Such an approach is not guaranteed to work, as the Lagrange multipliers of the stabilized model have no simple relation with those of the eddy-resolving model. The iterative search technique could then be led in the opposite direction as the true solution, as was shown to occur in a quasi-geostrophic ocean model (Köhl and Willebrand, 2002). We are aware of only one case where the unmodified method of Lagrange multipliers was applied to an eddy-permitting ocean GCM over a timescale longer than the eddy scale of a few months (Gebbie et al., 2006). Contrary to expectation given by the simple chaotic
models, an acceptable fit was found to oceanographic observations over a one-year interval in the Northeast Atlantic Ocean (Gebbie, 2007). No clear explanation for these disparate results has been put forward.

In this research, we wish to re-examine (2), the influence of nonlinear models on the method of Lagrange multipliers and ocean state estimation. Is the adjoint method useless with a highly-nonlinear or chaotic system, as studies with low-dimensional chaotic models suggest? Here we posit that the initialization problem that has informed much of the current thinking about the
Lagrange multiplier method is not the relevant analogy for ocean state estimation. The ocean state estimation problem may be





better described as a time-variable boundary value problem because synoptic atmospheric variability acts as an external forcing on the ocean (Section 2). Given our relatively uncertain knowledge regarding air-sea fluxes, the ocean state estimation problem should be considered a time-variable boundary value problem where both the initial conditions and boundary conditions must be found. In this case, the prospects for a successful state estimate are shown to be improved, even if one uses a highly

nonlinear model such as the forced, chaotic pendulum (Section 3). If the chaotic nature of the model is not a roadblock, what is the relevant criterion for success with the Lagrange multiplier method? Our results with the chaotic pendulum suggest that "controllability," defined as the ability to move from one arbitrary state to another by control adjustments, is the relevant parameter. The implications of these results is that there is a wide variety of situations where the Lagrange multipiers of an ocean general circulation model (GCM) are useful, and that previous GCM results can be explained in this context (Section 4).

## 2   Lagrange multiplier method

### 2.1   Pendulum model and synthetic data

The fixed, single pendulum can be modeled as a nonlinear or linear set of equations, and it can also be easily modified to be stable or unstable. In many ways, the pendulum is a more flexible and easily interpreted physical system than the often-used Lorenz (1963) equations that approximate atmospheric convection. The relevance of the pendulum to the ocean is obviously

indirect, but much of the community's knowledge of state estimation has been formed by the intuition of studies of simple models. The motion of the forced pendulum is described by (Baker and Gollub, 1990),

$$\frac{d^2\theta}{dt^2} + \frac{1}{q}\frac{d\theta}{dt} + \frac{g}{l}\sin\theta = f(t), \tag{1}$$

where $\theta$ is the displacement angle from vertical, $q$ is a damping coefficient, $g$ is gravitational acceleration, $l$ is the pendulum length, and $f(t) = b\cos(\omega_d t)$ is an external forcing term. With parameters $q = 100$ s, $g/l = 1.0\text{s}^{-2}$, $b = 1.5\text{rad s}^{-2}$, and

$\omega_d = 2/3$ s$^{-1}$, the pendulum is chaotic (here defined as extreme sensitivity to initial conditions). Following the numerical implementation in Appendix A, the state vector is defined, $\mathbf{x}(t) = [\omega(t)\theta(t)]^T$, where $^T$ is the vector transpose. Matrices and vectors are indicated in boldface. The state has dimension $M = 2$ and the forcing vector has dimension 1. The evolution of the state is succintly written,

$$\mathbf{x}(t + \Delta t) = \mathcal{L}[\mathbf{x}(t), f(t)], \tag{2}$$

where the model state is stepped from time $t$ to $t + \Delta t$, and $\mathcal{L}$ is the discretized, nonlinear operator that represents equation (1). In the ocean model case, the state would correspond to velocities and property fields, and the external forcing would include air-sea momentum, heat, and freshwater fluxes.

We consider an "identical twin" experiment where the true solution is known (solid line, Figure 1), and we observe the pendulum angle episodically through time with normally-distributed random errors of standard deviation, $\sigma_\theta = 0.5$ rad. In

most oceanographically-relevant cases, observations have already been collected over some fixed time interval ($0 \leq t \leq T$). Here, observations, $y(t)$, are taken at a set of $N_y$ evenly-spaced times with an time interval of $\Delta t_y = T/(N_y - 1)$.

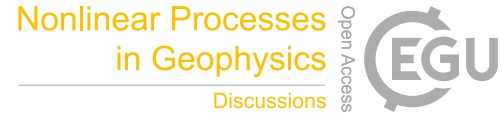

## 2.2 Cost function

We proceed by defining a least-squares cost function to be minimized. The data-based contribution to the cost function, $J_d$, measures the squared misfit between the model and observations:

$$J_d = \frac{1}{N_y} \sum_{i=0}^{N_y-1} \frac{[\theta(i\Delta t_y) - y(i\Delta t_y)]^2}{\sigma_\theta^2} \tag{3}$$

where the penalty is weighted by the number of observations, $N_y$, and their standard error, $\sigma_\theta$, such that the expected value of $J_d$ is near 1. As we have imposed Gaussian error statistics, minimizing this least-squares cost function also leads to the maximum likelihood solution (e.g., Jazwinski, 1970). In matrix-vector notation, equation (3) becomes,

$$J_d = \sum_{i=0}^{Ny-1} [\mathbf{E}\mathbf{x}(i\Delta t_y) - y(i\Delta t_y)]^T \, W^{-1} \, [\mathbf{E}\mathbf{x}(i\Delta t_y) - y(i\Delta t_y)], \tag{4}$$

where $y(t)$ is a scalar, $\mathbf{E}$ is the observational matrix that samples the observable part of the state and has dimension $1 \times M$,
and $W$ is a weight. Comparison of the first term in (4) to equation (3) shows that $W = N_y \sigma_\theta^2$. While it is unconventional to transpose the scalar data-model misfit in equation (4), we retain this notation so that the equations are applicable to cases where multiple observations are available at each time.

A second contribution to the cost function includes two terms that constrain the difference between our posterior and prior estimates of the initial conditions and forcing,

$$J_0 = [\mathbf{x}(0) - \mathbf{x}_0(0)]^T \mathbf{S}_x^{-1} [\mathbf{x}(0) - \mathbf{x}_0(0)] + \sum_{t=0}^{N_t-1} [f(i\Delta t) - f_0(i\Delta t)]^T S_f^{-1} [f(i\Delta t) - f_0(i\Delta t)] \tag{5}$$

where $\mathbf{x}_0(0)$ is the first-guess initial conditions, there are $N_t$ model timesteps, $f_0(t)$ is the first-guess forcing, and $\mathbf{S}_x$ and $S_f$ are weights that restrict the size of the perturbations to 5rad and 10rad s$^{-2}$, respectively. Here we seek values of $\mathbf{x}(t)$ and $f(t)$ that minimize the sum, $J' = J_d + J_0$, but the stationary point found by individually minimizing the values $dJ'/d\mathbf{x}(t)$ and $dJ'/df(t)$ will almost certainly violate the model constraint in equation (2). We enforce the model constraint by appending a
Lagrange multiplier term to the combined cost function,

$$J = J_d + J_0 - 2 \sum_{t=0}^{N_t-1} \mu(i\Delta t + \Delta t)^T \{\mathbf{x}(i\Delta t + \Delta t) - \mathcal{L}[\mathbf{x}(i\Delta t), f(i\Delta t)]\}, \tag{6}$$

where $\mu(t)$ is a Lagrange multiplier, and the scaling with "2" is helpful in later derivations and does not change the numerical value of $J$ because the quantity inside curly brackets vanishes. Now the cost function can be minimized by independently setting the partial derivatives of $J$ with respect to the state, the forcing, and the Lagrange multipliers to zero. This problem will
be solved using a gradient-descent method (detailed later) that is excellent at finding the nearest minimum. If the first guess is good, then the closest minimum may actually be the global minimum (e.g., Pires et al., 1996), and therefore we design an improved first guess next.



### 2.3 First-guess trajectory

Minimizing $J$ requires a first-guess of the full model trajectory, $\mathbf{x}_0(t)$. A sensible and common approach is to use the observation at initial time, $y(0)$, to inform the initial conditions for the state, $\mathbf{x}_0(0)$. Then, the first-guess forcing, $f_0(t)$, is used to drive the model forward in time. In this case, the state at any time, $\tau$, can be computed directly from the initial state,

$$\mathbf{x}_0(\tau) = \mathcal{L}_{K-1}[\ldots[\mathcal{L}_1[\mathcal{L}_0[\mathbf{x}_0(0), f_0(0)], f_0(\Delta t)]\ldots], f_0(K\Delta t - \Delta t)] = \mathcal{R}(\tau, 0)[\mathbf{x}_0(0)], \qquad (7)$$

where $\mathcal{L}_k$ indicates the nonlinear model operator at timestep $k$, $K = \tau/\Delta t$ is the number of timesteps between $t = 0$ and $t = \tau$, and the state transition matrix, $\mathcal{R}(m, n)$, defines the aggregate, nonlinear model step to time $m$ from $n$. In the following, we refer to this trajectory as the "standard" first-guess state.

For a nonlinear system, and a chaotic system in particular, this first-guess trajectory usually diverges from the already-collected observations at some point, and thus can be ruled out as a possible solution *a priori*. When the pendulum initial conditions are imperfectly known, the range of possible pendulum trajectories expands greatly with time, even if the forcing evolution is perfectly known (background shading, Figure 1). Normally-distributed initial perturbations to the truth with standard deviation of 1 rad s$^{-1}$ in angular velocity and 0.5 rad in the initial angle lead to a divergence of roughly 200 rad between extreme trajectories (background shading, Figure 1). The angle is not renormalized when the angle is greater or less than $\pi$, and thus the angle records a history of how many times the pendulum has rotated. If no information about the initial angular velocity is available, a reasonable assumption is that $\omega = 0$ with some large error, but the pendulum trajectory with this initial velocity and the correct initial angle diverges from truth in less than 5 seconds (first dashed line, Figure 1). In the case where the initial velocity is known perfectly but the initial angle is observed with an initial error of 0.5 rad (second dashed line, Figure 1), the trajectory follows truth for 30 seconds before eventually diverging. As the time interval of interest increases, any uncertainty in the initial conditions will ultimately lead to a divergence between truth and this first-guess model trajectory. While these sample model trajectories may seem overly naive, the first guess trajectory used for ocean state estimation usually has similar characteristics: usage of an observation at the initial time, some prior knowledge of the forcing, and a freely-running forward model.

### 2.4 An improved first guess

The aforementioned standard approach does not use the observational information already in hand that could inform the time evolution of the forcing. There are many methods that are available to update the forcing, such as the Kalman filter (e.g., Keppenne et al., 2005), but these methods rival or exceed the method of Lagrange multipliers in computational cost because of the explicit representation of the solution covariance matrix (Fukumori, 2002). Here we design a method that is computationally efficient and provides a good first guess for the boundary control problem.

Here we seek an update to the initial conditions and the forcing (i.e., $\omega_1(0) = \omega_0(0) + \delta\omega$, $\theta_1(0) = \theta_0(0) + \delta\theta$, $f_1(t) = f_0(t) + \delta f(t)$), that takes the observations into account. For small perturbations, we derive a linearized equation for the change





to the state at the time of the first observation, $t = \Delta t_y$,

$$\mathbf{x}_1(\Delta t_y) = \mathbf{x}_0(\Delta t_y) + [\Pi_{i=1}^{K}\mathbf{A}((K-i)\Delta t) \,|\, \Pi_{i=1}^{K-1}\mathbf{A}((K-i)\Delta t)\mathbf{B} \,|\, \Pi_{i=1}^{K-2}\mathbf{A}((K-i)\Delta t)\mathbf{B} \,|\ldots|\, \mathbf{B}] \begin{pmatrix} \delta\omega(0) \\ \delta\theta(0) \\ \delta f(0) \\ \delta f(\Delta t) \\ \vdots \\ \delta f(\Delta t_y - \Delta t) \end{pmatrix} + \epsilon \tag{8}$$

where $\mathbf{x}_1$ is the "improved" first-guess, $K = \Delta t_y/\Delta t$ is the number of model time steps from $t = 0$ to $t = t_y$, $\mathbf{A}(t) = \partial\mathcal{L}/\partial\mathbf{x}(t)$ is the tangent-linear model, $\mathbf{B} = \partial\mathcal{L}/\partial f(t)$ is constant in time, and $\epsilon$ is the error due to linearization. We define the column

vector of perturbations in equation (8) to be the control vector, $\mathbf{u}$, so that the equation becomes:

$$\mathbf{x}_1(\Delta t_y) = \mathbf{x}_0(\Delta t_y) + \mathbf{C}\mathbf{u} + \epsilon, \tag{9}$$

where $\mathbf{C}$ is the controllability (or reachability) matrix (e.g., Dahleh and Diaz-Bobillo, 1999; Wunsch, 2010).

The observation, $y(\Delta t_y)$, and the combination of equations (7) and (9) provides one constraint,

$$y(\Delta t_y) = \mathbf{E}\mathcal{R}(\Delta t_y, 0)[\mathbf{x}_0(0)] + \mathbf{E}\mathbf{C}\mathbf{u} + \mathbf{n} \tag{10}$$

where the controllability matrix can be calculated given the trajectory, $\mathbf{x}_0(t)$, and $\mathbf{n}$ is the misfit. Here we minimize the squared misfit,

$$J_1 = \{y(\Delta t_y) - \mathbf{E}\mathcal{R}(\Delta t_y, 0)[\mathbf{x}_0(0)] - \mathbf{E}\mathbf{C}\mathbf{u}\}^T W^{-1}\{y(\Delta t_y) - \mathbf{E}\mathcal{R}(\Delta t_y, 0)[\mathbf{x}_0(0)] - \mathbf{E}\mathbf{C}\mathbf{u}\} + \mathbf{u}^T \mathbf{Q}^{-1}\mathbf{u}, \tag{11}$$

where $\mathbf{Q}$ is a block diagonal matrix with $\mathbf{S}_x$ and $S_f$ on the diagonal. We solve with the method of total inversion (Tarantola and Valette, 1982),

$$\mathbf{u} = W\mathbf{C}^T\mathbf{E}^T[\mathbf{E}\mathbf{C}W\mathbf{C}^T\mathbf{E}^T + \mathbf{Q}]^{-1}\{y(\Delta t_y) - \mathbf{E}\mathcal{R}(\Delta t_y, 0)[\mathbf{x}_0(0)]\}, \tag{12}$$

where we update the state transition and controllability matrices iteratively. The full nonlinear model is run with the updated controls to produce the improved first-guess trajectory, $\mathbf{x}_1(t)$. The algorithm proceeds sequentially $N_y - 1$ times, where the terminal state from one segment becomes the initial condition for the next.

## 2.5 Solution for Lagrange multipliers

We obtain the sensitivity of $J$ to the initial conditions by taking the partial derivative,

$$\frac{\partial J}{\partial\mathbf{x}(0)} = 2\mu(\Delta t) + 2\mathbf{S}_x^{-1}[\mathbf{x}_1(0) - \mathbf{x}_0(0)] + 2\mathbf{E}^T W^{-1}[\mathbf{E}\mathbf{x}_1(0) - y(0)], \tag{13}$$



where the improved first guess, $\mathbf{x}_1(t)$, is used. Taking the derivative with respect to the other set of unknowns, we find,

$$\frac{\partial J}{\partial f(t)} = 2\mathbf{B}^T \mu(t+\Delta t) + 2S_f^{-1}[f_1(t) - f_0(t)]. \tag{14}$$

With knowledge of these gradients, we could improve the initial conditions and forcing, but both equations (13) and (14) depend upon the Lagrange multipliers, $\mu(t)$, that we must solve for first.

Extending the partial derivative of $J$ with respect to $\mathbf{x}(t)$ and $\mu(t)$ at all times, we recover the Euler-Lagrange (or "normal") equations. The Lagrange multipliers are determined by timestepping backward in time:

$$\mu(t) = \mathbf{A}(t)^T \mu(t+\Delta t) + \mathbf{E}^T W^{-1}\left[\mathbf{E}\mathbf{x}_1(t) - y(t)\right], \tag{15}$$

where the last term on the right hand side only appears if an observation, $y(t)$, is available. Initial conditions are given by,

$$\mu(T) = \mathbf{E}^T W^{-1}\left[\mathbf{E}\mathbf{x}_1(T) - y(T)\right]. \tag{16}$$

Equations (15) and (16) are collectively known as the *adjoint model* (e.g., Bugnion et al., 2006), where the model-observation misfit is part of the adjoint model forcing, $\mathbf{E}^T W^{-1}[\mathbf{E}\mathbf{x}_1(t) - y(t)]$. Now the result of equation (15) can be substituted into equations (13) and (14) to solve for the gradients.

     In summary, the method of Lagrange multipliers is implemented with the following steps. Starting with a guess for the initial conditions and forcing, $\mathbf{x}_0(0)$ and $f_0(t)$, we improve the first guess and solve for the full trajectory, $\mathbf{x}_1(t)$, using the method
of total inversion. The adjoint model is then run backward in time to solve for the sensitivity of $J$ with respect to the two types of unknowns, $\mathbf{x}(0)$ and $f(t)$. Here we use a quasi-Newton gradient descent optimization (Nocedal, 1980) to update these uncertain control parameters. Because the model is nonlinear, the tangent-linear model, $\mathbf{A}(t)$, will depend upon the nonlinear model trajectory, and we re-run the full nonlinear model to get an updated trajectory that will replace $\mathbf{x}_1(t)$. Forward-adjoint model integrations are repeated until $J$ has an acceptable value by a $\chi^2$ statistical test.

## 3   Results

### 3.1   Tracking chaotic transitions

Synthetic observations of the pendulum angle are generated every 2.5 seconds over a 50 second time interval, where a random error of 0.5 rad is added to every observation. We first illustrate the futility of a brute force search for the optimal initial conditions by re-running the forward model with combinations of the initial angle and angular velocity in the neighborhood
of the truth. The data contribution to the cost function (equation 3) is then evaluated for each forward model trajectory, giving rise to a complex topology where the global minimum is not immediately visible (Figure 2). That the topology of the cost function is not conducive for gradient descent search was previously documented in models of convection, quasi-geostrophic flow, and the oceanic double gyre model (e.g., Miller et al., 1994b; Köhl and Willebrand, 2003; Lea et al., 2006). When the initial angle and angular velocity are slightly perturbed from the truth, the cost function values can become extremely large due
to the divergence of trajectories. Furthermore, the cost function varies irregularly with many local extrema at locations other





than the true solution. The basin of attraction of the true solution, defined in analogy to a drainage basin on a topographic map, is much smaller than the observational uncertainty, and thus, it is likely that the iterations of the adjoint method will converge to a local, non-global, minimum.

We start the application of the methods of Section 2 by determining the first-guess trajectory. Here we implement a model
timestep of $0.01$ s over a $50$s integration time and thus the control vector has $5000$ forcing variables and $2$ initial condition variables. The first-guess initial conditions, forcing, and trajectory are calculated according to Section 2.3. Despite the assumption of linearity in the calculation of the first guess, the improved first guess has a trajectory that is nearly consistent with the error bars of the observations (gray line, top panel, Figure 3). The first guess also tracks the rapid transitions in the interval, $12$s$< t < 20$s, where revolutions of the pendulum occur due to chaotic dynamics. These results contrast with a seemingly
reasonable first-guess trajectory that is determined by a model simulation initialized with the first observation ($\theta(0) = y(0)$) and zero angular velocity that goes off track in less than 5 seconds (see "standard first guess", top panel, Figure 3). Similar first-guesses are common in ocean models, where the circulation field is started from rest with the assumption that geostrophic balance will equilibrate the velocity field rapidly. Our more sophisticated, but still linear, method of deriving an improved first guess makes the Lagrange multiplier method more likely to succeed.

The improved first-guess trajectory is a better fit to the data in large part due to the updated initial angular velocity (middle panel, Figure 3). Starting with an angular velocity of about 2 rad/s, this trajectory has frequent changes in the sign of angular velocity consistent with reversals in pendulum rotation. Conversely, the standard trajectory has a long period of strictly positive angular velocity ($10$ s$< t < 40$ s) that is inconsistent with the observations. Another important factor is the position of the pendulum around 10 seconds after the start of the integration, when small differences in the state become greatly magnified.
The true pendulum trajectory then enters a period where several revolutions occur successively. The inaccurate initial velocity causes errors at this critical time of instability and thus the trajectories diverge.

For $\theta$ and $\omega$, the difference between the improved-first-guess and final (Lagrange multiplier method) trajectories is smaller than the changes brought about by the first-guess improvement itself. The method of Lagrange multipliers acts similarly to a combined filter-smoother that simultaneously takes into account past and future observations, leading to an angular velocity
evolution with somewhat smaller range while still fitting the observations. Consequently, the evolution of the pendulum angle is also smoother, with fewer variations at the observational sampling frequency of $1/2.5$s. The full impact of the Lagrange multiplier method only becomes clear when considering the external forcing in the following section.

### 3.2 Reconstruction of the forcing

The improved first-guess trajectory better fits the observations than a standard first-guess, but there are tradeoffs in the estimated
forcing (bottom panel, Figure 3). In order to track the chaotic transitions of the pendulum, the improved first-guess trajectory makes forcing adjustments that are sometimes strong and abrupt in order to compensate for previous errors in the trajectory. In other words, these adjustments take the forcing evolution farther away from the truth that was used in the standard first-guess trajectory. This is reminiscent of the small-scale features that are added to the surface forcing of ocean models in order to fit observations (e.g., Stammer et al., 2002b), although our case is not a compensation for inaccurate model dynamics because





we are operating under a perfect model assumption. This tradeoff is probably unacceptable for those wishing to physically interpret the forcing field, and indicates that the improved first-guess estimate is not a good final solution despite fitting the observations. The power of the Lagrange multiplier method is now clear; not only is the final estimate smoother than the first guess, the final forcing estimate accurately reproduces the amplitude and frequency of the true forcing: $f(t) = 1.5 \cos(2t/3)$.

5 Our identical twin experiment permits a comparison with the truth to diagnose actual errors even at times without observations. While the improved first guess appeared to fit the data well, the misfit to the truth displays considerable structure, including a large deviation around $t = 45$s to values that are inconsistent with the observations (top panel, Figure 4). Such a deviation reflects an inaccurate interpolation between data points during the construction of the improved first guess trajectory. The first guess also appears to overfit the observations, as this estimate deviates from the truth in the neighborhood of observations with large error (e.g., $t = 32.5, 42.5$s). The final estimate, on the other hand, hovers near 0 for the entire time interval, with a standard deviation of 0.46 rad, very near to the actual observational uncertainty of 0.50 rad. The final estimate reproduces 72% of the variance in the observational error, computed by comparison of the estimated to true observational error. Visually, the final estimate is closer to the truth than the observations over the majority of the time interval, indicating that the Lagrange multiplier method filters out the observational noise even in this chaotic system.

15 For the angular velocity and forcing (middle and bottom panels, Figure 4), the Lagrange multiplier method reproduces the truth despite the imperfect, sparse observations and the chaotic model dynamics. The suppression of the abrupt and large changes in forcing is not simply a smoothing or averaging of the forcing, but instead is seen to reflect the true forcing, as evidenced by the deviation from true forcing being small. The method of Lagrange multipliers is therefore superior to the first-guess estimate because all components of the solution, both the state and forcing, can be physically interpreted.

20 ## 3.3 Influence of the data stream

In the case where only two observations are available ($\theta(0) = -2 \pm 0.5, \theta(50) = 10 \pm 0.5$), the time between observations is greater than both the fundamental frequency and the nonlinear timescale of the pendulum. The nonlinear timescale is here defined as the time interval that the tangent linear model well-approximates the nonlinear dynamics, which depends upon the size of initial perturbation. Despite the long time between observations, the estimated trajectory fits both observations via the Lagrange multiplier method (top left panel, Figure 5). Thus, there appears to be no lower limit on the number of observations necessary in order to produce an acceptable state estimate with this model. Of the two steps in our method, it is the first-guess calculation that is responsible for fitting the data within their errors, and the optimization with Lagrange multipliers doesn't substantially change or improve this estimate. When comparing the estimate to the true trajectory that was withheld from the reconstruction method, differences larger than 10 rad exist (bottom left panel, Figure 5). Thus, reconstruction of the full, partially-unobserved trajectory without any intervening observations is a challenging task, as expected. Here, we emphasize that the first goal in state estimation is to find any model trajectory that fits the observations, and that in realistic cases we will not know whether the model interpolates between the observations in the correct way. We recognize that a chaotic model usually has many trajectories that satisfy the initial and final times, and thus, any one trajectory is unlikely to reconstruct the truth at all intervening times.





In the case that many ($N_y = 20$) observations are taken but the standard error is large (5 rad), all observations are again fit within their $1\sigma$ error bars. Strictly speaking, the data appear to be overfit, as $32\%$ of the points are expected to reside outside the one standard error level but none do. This fit has a lower standard deviation, 3.3 rad, than that expected by the observational error of 5.0 rad, suggesting that the numerical model is adding information that is complementary to the observations. Only in short time intervals does the estimate differ from the truth by more than 5 rad, such as near the observation of the $2\sigma$ outlier at $t = 25$s. Unlike the case where 20 observations were taken with a smaller standard error of $0.5$ rad in the previous section, this estimate is only partially successful at filtering noise out of the observations. The correlation coefficient between the estimated and actual observational error is $r = 0.43$, indicating that some noise remains. Both case studies in this section indicate that neither the quality nor the quantity of the data stream affect whether the Lagrange multiplier method can be successful with a chaotic model.

### 3.4 Influence of the number of controls

The previous section addresses cases where the forcing is adjusted at every timestep, leading to 5002 control variables. After application of the Lagrange multiplier method, the resulting value of the data-based component of the cost function (equation 3) depends upon both the number of control parameters and the number of observations (Figure 6). The value of $J_d$ is always below 1 when 5002 control variables are defined, consistent with the previously-reported results. Here we test the null hypothesis that the model is consistent with the observations, and we perform a $\chi^2$ statistical test with $N_y$ degrees of freedom as is appropriate for independent observations. Our one-sided test statistic is the value of $J_d$ where $5\%$ of cases are expected to have larger values by chance. For 5002 controls, we find that all values are small enough that the null hypothesis cannot be rejected at the $5\%$ insignificance level (area above black line, Figure 6). Thus, the Lagrange multiplier method is expected to acceptably fit the observations if enough controls are available. In the ocean state estimation problem, all air-sea fluxes are uncertain and temporally variable so there is no shortage of controls that can be defined.

For a given number of observations, a decrease in the number of controls leads to a decrease in the likelihood of a successful fit to the data. The initialization problem is equivalent to the case with two boundary control variables, and Figure 6 suggests that the Lagrange multiplier method will not produce a good fit to data, as documented by previous works. The criterion of a good fit also depends upon the number of observations, where more observations decrease the likelihood of success. To understand why the data can or cannot be fit, consider that each observation gives a constraint of the type documented in equation (10). If all of these constraints are enforced simultaneously, the problem is formally underdetermined when the number of controls exceeds the number of observations, and it is generally likely that a solution exists. The simple interpretation that the number of controls must exceed the number of observations does not strictly hold due to the logarithmic scale in Figure 6. Even when the problem is formally underdetermined, a singular value decomposition analysis of the controllability matrix, $\mathbf{C}$, reveals that not all controls are independent and that the data cannot always be fit perfectly. Here, we identify formally underdetermined cases with a $J_d$ value that is unacceptably high (Figure 6), thus the solvability condition is sometimes violated. Such a result indicates that the controllability matrix has an effective rank less than the number of observations, showing the importance of




this quantity as a diagnostic measure of the conditioning of the estimation problem. In practice, the singular values need not be strictly zero, as a large discrepancy between the magnitudes of singular values can give ill-conditioning.

We also find cases where the gradient descent method is capable of navigating the complex cost function topology with Lagrange multipier sensitivity information. A slice of $J$ along $\omega(0)$ and $\theta(0)$ is focused on a region of phase space that appears

to contain a local minimum, although this is not the true solution (left panel, Figure 7). In the initial control problem, the optimization would proceed in these two dimensions and be trapped by the local minimum. Taking a 2D slice of the cost function in the dimension of the initial angular velocity and forcing, $\delta f(0)$, however, the same location may no longer be a local minimum in the expanded phase space (right panel, Figure 7). In our example, the cost function can be further minimized by decreasing $\delta f(0)$, and the optimization process may eventually get out of the trap in $\omega(0)/\theta(0)$ space. Thus, additional

dimensions in the optimization space can sometimes alleviate problems with the gradient descent algorithm.

## 4 Discussion

### 4.1 Relation to Kalman Filter/Smoother

Our suggestion that controllability is a key criterion brings our understanding of the Lagrange multiplier method into closer consistency with the Kalman filter and smoother. Both methods solve the same least-squares problem, and the solution of a

linear problem should not depend upon the chosen method (e.g., Wunsch, 2010). Fukumori et al. (1993) found that the problem must be controllable for the Kalman filter/smoother to be successful. In addition, the chaotic Lorenz (1963) model was tracked with the Kalman filter/smoother over time windows much longer than the nonlinear timescale when the system was completely controllable (i.e., all estimated quantities are uncertain and are treated as control variables) (Evensen, 1997). Our results suggest that the equivalence of the Kalman filter/smoother and Lagrange multiplier method may be extended to nonlinear problems,

thus explaining why the chaotic estimation problem may be solved by the Lagrange multiplier method.

To recover the true trajectory of a system, observability is also important (Fukumori et al., 1993; Marchal, 2014). Wunsch (1996) states that, "Problems owing to the multiple minima in the cost function can always be overcome by having enough observations to keep the estimates close to the true state." To evaluate this statement, we emphasize that there are two levels of successful reconstruction: 1) one that accurately fits the data, and 2) one that accurately fits the truth at all times and locations.

Criterion 1) has been our metric for success in this work, as in real-world problems, criterion 2) cannot be tested. Here we have shown that only controllability is necessary for 1), even with a nonlinear system. In addition, we show that the data can still be fit even if very few observations are available, as an off-track estimate can be righted by precise adjustments to the forcing (recall Figure 3). That short-lived forcing adjustments can put the estimate on track is likely a consequence of the nonlinear dynamics of our particular problem, although we believe that an eddy-resolving ocean general circulation model could behave

the same way. Our interpretation is consistent with work in the control of chaotic systems. Engineers have described the control of a chaotic system as being "easier" than control of other systems because the necessary control adjustments are very small (e.g., Ott et al., 1990).



## 4.2 Comparison of controllability and stability metrics

In the section, we compare criteria for the success of the Lagrange multiplier method. Previously-suggested criteria include Lyapunov exponents or other stability metrics of the tangent-linear model (e.g., Lea et al., 2000). The tangent linear matrix has eigenvalues with absolute value greater than one when linearized about a state in the upper-half plane ($\pi/2 < \theta < 3\pi/2$),

reflecting the divergence of neighboring nonlinear trajectories when a pendulum perturbed towards the horizontal is more rapidly accelerated downwards. Conversely, the lower-half plane is locally linearly stable. The unforced pendulum with initial conditions in the upper-half plane is episodically-unstable, until damping brings the pendulum permanantly into a stable configuration in the lower half-plane.

Here we investigate the influence of stability versus that of nonlinearity. The pendulum is a useful system because it is easily

modified to have 4 distinct dynamical states: 1) nonlinear, unstable, 2) nonlinear, stable, 3) linear, stable, and 4) linear, unstable. Case 1) is the original dynamical equation for the pendulum (equation 1). By restricting the phase space to the lower-half plane, the pendulum is locally linearly stable at all times, although it is still nonlinear (Case 2). When the pendulum is linearized by the small-angle approximation with the linear term of the Taylor series expansion ($\sin \theta \approx \theta$, see Appendix A2), we obtain a linear, stable model (Case 3). If instead the pendulum is linearized around its apex, the sign of $\theta$ in the linearized equation is

reversed, rendering the system linear but unstable (Case 4).

We revisit the problem of estimating the initial angle when $\theta$ is observed at the final time. A synthetic observation is generated by running the model with initial displacement of $-\pi/6$ rad and zero velocity. Assuming a perfect model and observation, the shape of the cost function is generated by changing the initial conditions and evaluating $J$. In the two nonlinear cases, a slice of the cost function contains many local minima that emerge when the time window is extended from 5 to 50 seconds (Figure 8,

*lower panels*). The cost function in the nonlinear, stable case deviates from a parabola because the state transition matrix is non-self-adjoint and non-normal growth occurs (Farrell 1989; Farrell and Moore 1993). Thus, even nonlinear models that are stable are subject to local minima, and linear stability is not always a good metric to determine whether a gradient descent search will successfully find the global minimum.

Conversely, the linear, unstable case does yield a parabolic cost function (Figure 8, *upper panels*), implying that instability

does not impede the search for the minimum. Again, local linear stability doesn't appear to be a good metric for determining the presence of local minima, because an unstable system may yield a well-behaved function. This example reinforces the counterintuitive relationship between stability and local minima, where a linearly stable system does not have a paraboloidal cost function but an unstable system does. While this reversed relationship does not always hold, linear stability metrics are not reliable. We suggest that controllability is a better metric, but note that controllability and stability are not unrelated, as a

system with a growing unstable mode could lead to a controllability matrix that effectively drops rank.

## 4.3 Relevance to ocean state estimation

In the Introduction, we remarked on the only ocean state estimate known to the authors that successfully implemented the Lagrange multiplier method in an eddy-permitting ocean GCM without any modification to the adjoint model (Gebbie et al.,



2006). In light of the results of this work, a combination of factors appears to have been responsible for that success. While the ocean model had $1/6° × 1/6°$ horizontal resolution and contained mesoscale eddies (Figure 9), the resolution was not adequate to fully resolve the eddies. Also, the model domain of the Northeast Atlantic Ocean was a relatively quiescent one. Both factors likely led to the ocean model being more linear than other studies. In addition, the adjoint model of a coarse-resolution twin

was used to form an improved first guess, which would improve the likelihood of success with gradient descent much as our method did here. Perhaps most importantly, the ocean state estimate included air-sea control fields that were updated every 10 days, leading to a total of $5.5 × 10^6$ control variables. Given the rapid adjustment of the ocean to barotropic waves, it is likely that the system passed the controllability criterion derived in this work. Controllability could be numerically evaluated in a GCM by a series of impulse functions: a dynamical equivalent to the passive response recorded by transit time distributions

(e.g., Delhez et al., 1999; Haine and Hall, 2002). Open questions include whether the deep ocean is completely controllable by surface boundary conditions, and whether ocean data require variability at timescales shorter than 10 days to be introduced through the surface forcing.

## 5   Conclusions

There is no fundamental obstacle to constraining a highly nonlinear model to observations using the Lagrange multiplier

method. On the basis of research primarily with toy models, chaotic systems were thought to represent such an obstacle if the estimation time window was too long. Here we find that the trajectories of the nonlinear pendulum can be tracked over multiple rapid transitions that are due to chaotic dynamics. The Lagrange multiplier method is successful under the condition that enough boundary controls are available through time, and that the system passes a test of controllability. In the case of the pendulum, the rank of the controllability matrix is a better metric to predict a success of state estimation rather than a measure

of dynamical stability. The ocean state estimation problem is analogous to the problem posed here; uncertain air-sea fluxes contain large errors that require control adjustments through time.

Our implementation of the Lagrange multiplier method includes a step to construct a good first guess that helps the iterative gradient descent search. The first-guess method has been developed with implementation in an ocean GCM in mind. Specifically, sub-problems are defined over the interval between observations and thus require less memory than a whole-domain

approach. In addition, we suggest that the particular first-guess method of this work is not the only way to produce a good first guess, and that other methods would bring the first-guess state close enough to the truth to increase the likelihood of success. A good example is the Green's function method (e.g., Stammer and Wunsch, 1996; Menemenlis et al., 2004) that selects a subset of the full control variables and makes some linearity assumptions. Following up the Green's function optimization with a gradient descent search with the Lagrange multiplier method is therefore a worthwhile research goal. The results of this work

suggest that ocean state estimation should continue with the Lagrange multiplier method and models that resolve higher and higher resolution physics.



## Appendix A: Numerical implementation of pendulum

### A1 Nonlinear pendulum

The forced, nonlinear pendulum is governed by the following equation,

$$\frac{d^2\theta}{dt^2} + \frac{1}{q}\frac{d\theta}{dt} + \frac{g}{l}\sin\theta = f(t), \tag{A1}$$

where the symbols were defined in Section 2. Discretizing in time, we obtain the symbolic form of the model equation used in the main text:

$$\frac{d\mathbf{x}(t)}{dt} = \frac{d}{dt}\begin{pmatrix} \omega(t) \\ \theta(t) \end{pmatrix} = \begin{pmatrix} -\frac{1}{q}\omega(t) - \frac{g}{l}\sin\theta(t) + f(t) \\ \omega(t) \end{pmatrix}. \tag{A2}$$

If the system is discretized with a forward Euler timestep of time $\Delta t$, the discrete-time state space realization is:

$$\begin{pmatrix} \omega(t+1) \\ \theta(t+1) \end{pmatrix} = \begin{pmatrix} (1-\Delta t/q)\,\omega(t) - (g\sin\theta(t)/l + f(t))\Delta t \\ \Delta t\,\omega(t) + \theta(t) \end{pmatrix}, \tag{A3}$$

or simply,

$$\mathbf{x}(t+1) = \mathcal{L}[\mathbf{x}(t), f(t)], \tag{A4}$$

where $\mathcal{L}$ is a nonlinear operator due to the sine function. For use in the Euler-Lagrange equations, we also produce the following linearized operators:

$$\frac{\partial\mathcal{L}}{\partial\mathbf{x}(t)} \equiv \mathbf{A}(t) = \begin{pmatrix} 1-\Delta t/q & -g\cos\theta(t)\Delta t/l \\ \Delta t & 1 \end{pmatrix}, \frac{\partial\mathcal{L}}{\partial f(t)} \equiv \mathbf{B} = \begin{pmatrix} \Delta t \\ 0 \end{pmatrix}. \tag{A5}$$

Here we use a second-order Taylor timestepping (i.e., midpoint forward Euler method) for increased accuracy. We code the tangent-linear model in accordance with differentiation rules for numerical codes (Griewank, 2000), and we run this linearized model with perturbations to all elements of $\delta\mathbf{x}(t)$ and $\delta f(t)$ to recover the values. Numerical parameters include: $\Delta t = 0.01$s, $\omega_{true}(0) = 1.2959$rad/s, $\theta_{true}(0) = -2.4667$rad, and the forcing phase, $\phi_{true}(0) = 0.3412$rad.

### A2 Linear, stable pendulum

The linear, stable pendulum is derived with the small-angle approximation. This approximation is a linearization around zero displacement

$$\begin{pmatrix} \omega(t+\Delta t) \\ \theta(t+\Delta t) \end{pmatrix} = \begin{pmatrix} 1-q\Delta t & -g\Delta t/l \\ \Delta t & 1 \end{pmatrix} \begin{pmatrix} \omega(t) \\ \theta(t) \end{pmatrix}. \tag{A6}$$

The more general tangent linear model is re-linearized around a changing nonlinear model trajectory.





*Acknowledgements.* We thank Geir Evensen, Armin Köhl, Olivier Marchal, and Eli Tziperman for discussions on this topic over the last decade, and to Jacques Verron for his note that has encouraged this work. GG also acknowledges Carl Wunsch for his guidance on originally selecting this project. GG was funded through the Ocean and Climate Change Institute of the Woods Hole Oceanographic Institution. TLH was funded by the Woods Hole Oceanographic Institution Summer Student Fellowship program through the U.S. National Science Foundation.

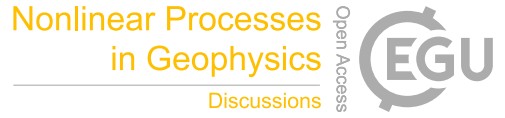

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

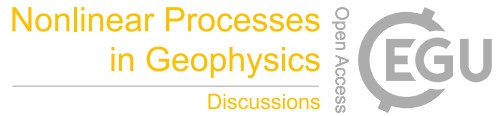

**Figure 1.** The rapid divergence of pendulum trajectories is indicated by the path density of trajectories (background shading), and the evolution of 3 sample trajectories: the "truth" or reference trajectory (*solid line*), a "first-guess" trajectory with incorrect initial angular velocity that diverges within $5s$ (*dashed line*), and a first-guess trajectory with incorrect initial angle, $\theta$, that diverges after $30s$ (*other dashed line*). The path density of trajectories is computed with 10,000 forward integrations with normally-distributed perturbations about the truth (standard deviation: $\sigma_\omega = 1$ rad/s, $\sigma_\theta = 0.5$ rad.





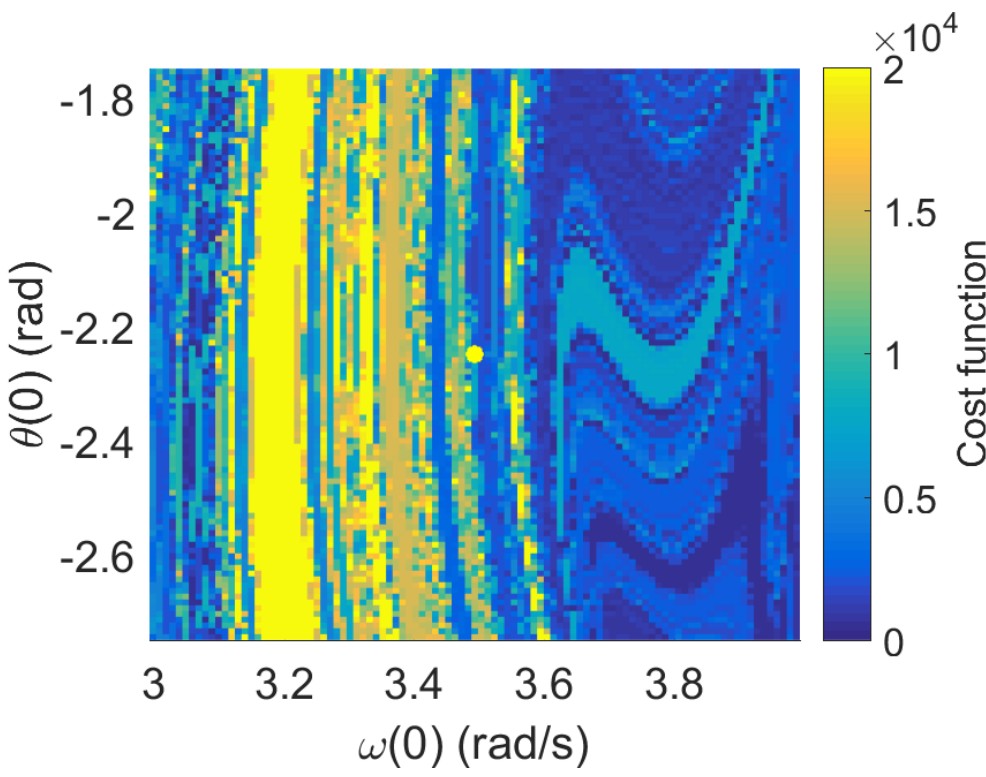

**Figure 2.** Cost function values as a function of initial angle, $\theta$, and angular velocity, $\omega$, for an observational time window of 50s. The minimum in this range of phase space (*yellow dot*) would not be identifiable by eye. This figure was produced with the parameters, $T = 30$s, $\Delta t_y = 1$s, and $q = 4$ s.





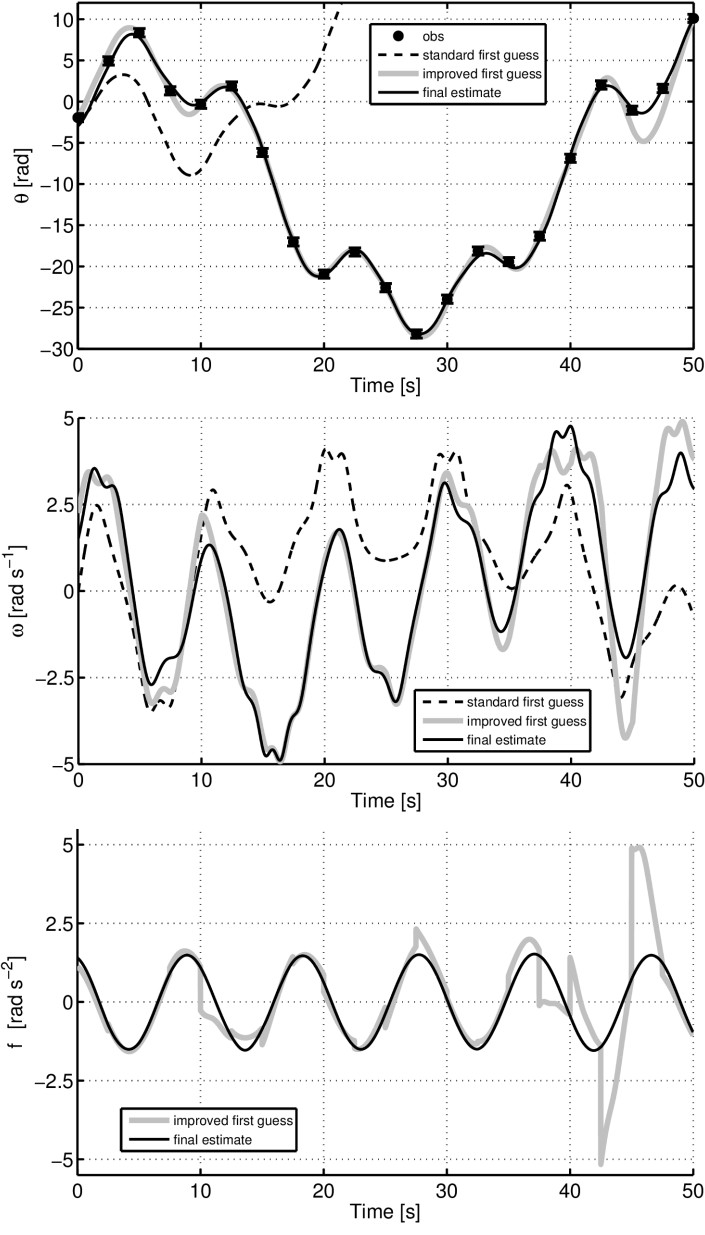

**Figure 3.** Control of the chaotic pendulum with an improved first guess and the Lagrange multiplier method. *Top panel*: Observations are taken every 2.5s with standard error of 0.5 rad (*circles with 1σ error bars*). The trajectory of the pendulum angle ($\theta(t)$, *top panel*), angular velocity ($\omega(t)$, *middle panel*), and the forcing ($f(t)$, *bottom panel*) are given for a standard first guess (*dashed line*), the improved first guess (*gray line*), and the final Lagrange multiplier-based estimate (*black line*). The standard first-guess forcing is not shown due to its similarity to the final estimate.





**Figure 4.** Comparison of the reconstructed pendulum and truth. *Top*: Difference between the observed (*circles*), standard first guess (*dashed*), improved first guess (*solid gray line*), and final estimate (*solid, black line*) of pendulum angle relative to the truth. The standard first-guess is off-scale for much of the panel. *Middle*: Similar but for angular velocity, $\omega$. *Bottom*: Same but for forcing, $f$. The standard first guess forcing is suppressed because it is identically zero.





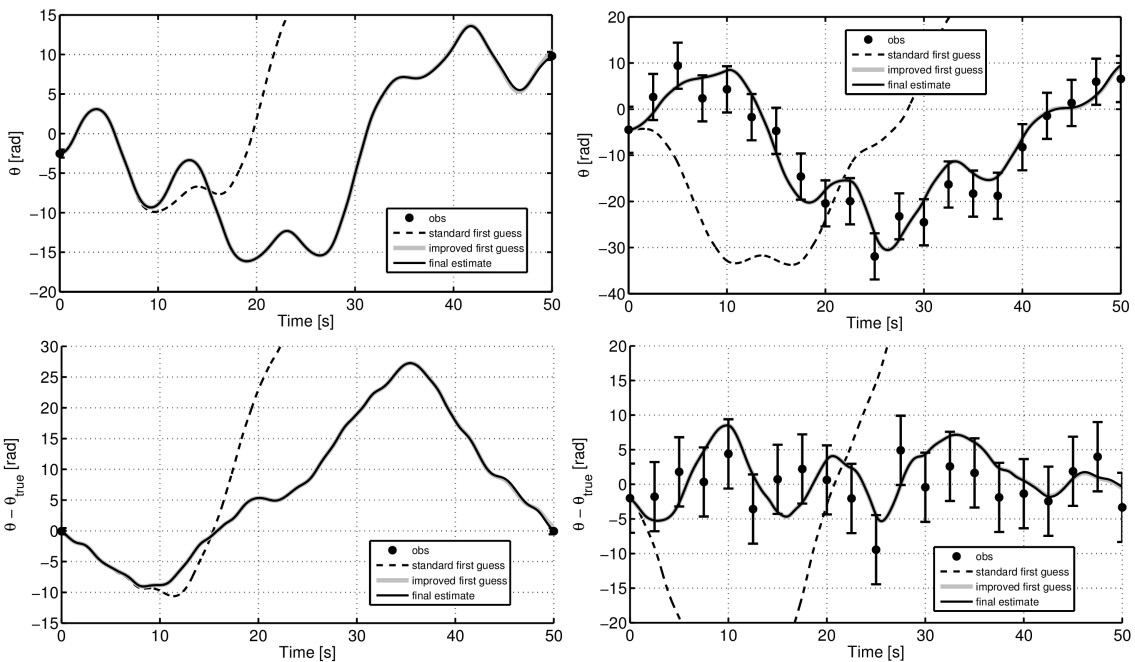

**Figure 5.** State estimation of the chaotic pendulum with a reduced set of 2 observations with standard error 0.5rad (*left column*) and a set of 20 observations with standard error 5rad. *Top*: Comparison of the observations (*circles with* $1\sigma$ *error bars*), the standard first guess (*dashed*), the improved first guess (*gray solid line*), and the final state estimate (*solid black line*), as in Figure 3. *Bottom*: Similar to the top row, except all quantities are referenced to the truth, $\theta_{true}$, as in Figure 4. Again the standard first guess is offscale for much of the time window. The improved first guess is nearly identical (and obscured) by the final estimate in all panels.





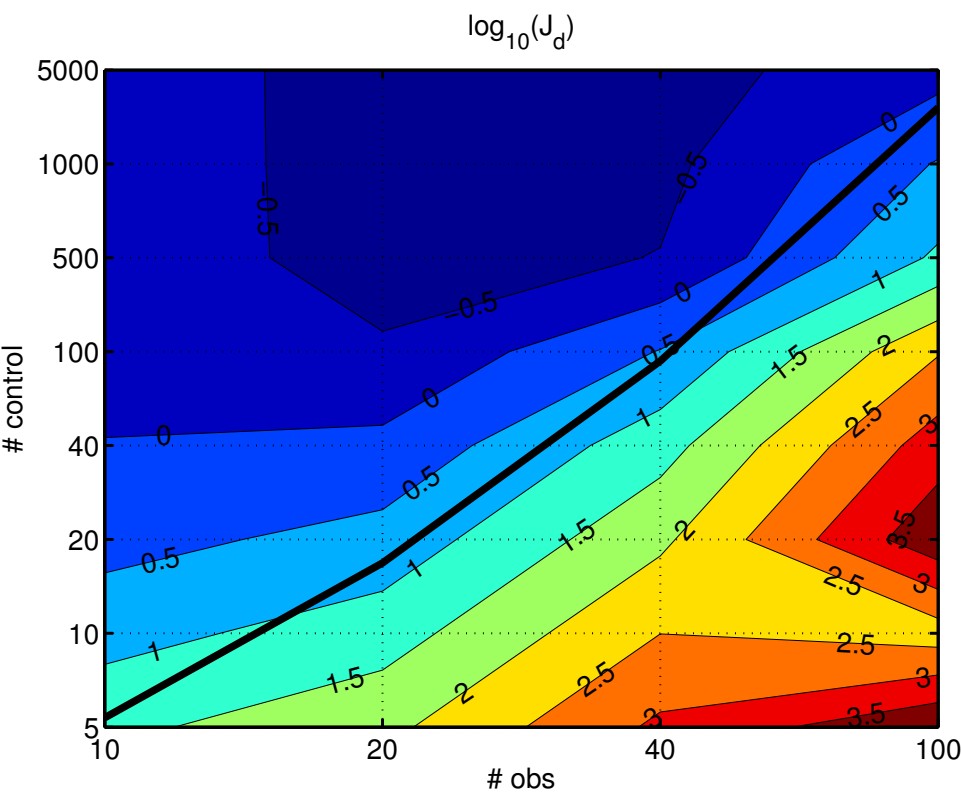

**Figure 6.** Influence of the number of observations and controls on the ability to track the chaotic pendulum. The base-10 logarithm of the data component of the cost function, $\log_{10} J_d$, is calculated as a function of the number of evenly-spaced observations over a 50 second window (*the abscissa*), and the number of effective degrees of freedom in the control perturbations to the forcing (*ordinate*). A $\chi^2$ statistical test determines the limit where 95% of realizations are expected to have smaller $J_d$ values (*thick, black line*); therefore, cases below this threshold represent an unacceptable fit to the data at the 5% insignificance level.

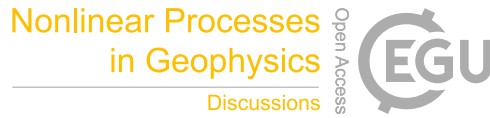



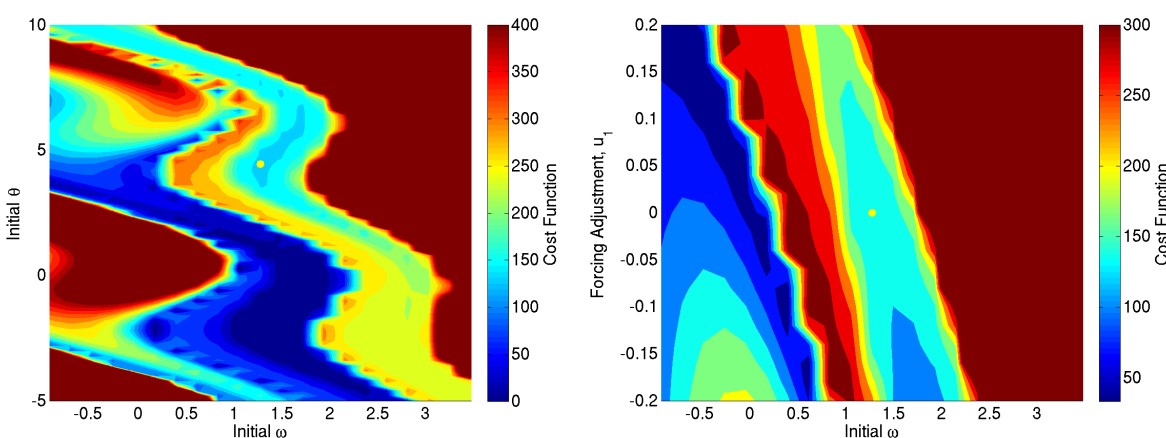

**Figure 7.** Escaping an apparent local minimum. *Left*: Cost function values as a function of $\theta(0)$ and $\omega(0)$, in a region of phas space where a local minimum is present in this slice (*yellow dot*). The same cost function, but oriented along a slice with constant $\theta(0) = 0$ in the dimensions of $u_1 = \delta f(0)$ and $\omega(0)$. The local minimum is the first two dimensions is no longer an extremum in the other two dimensions or the combined three-dimensional space. This case used the parameters, $T = 10\text{s}$ and $\Delta t_y = 1\text{s}$, for illustration.





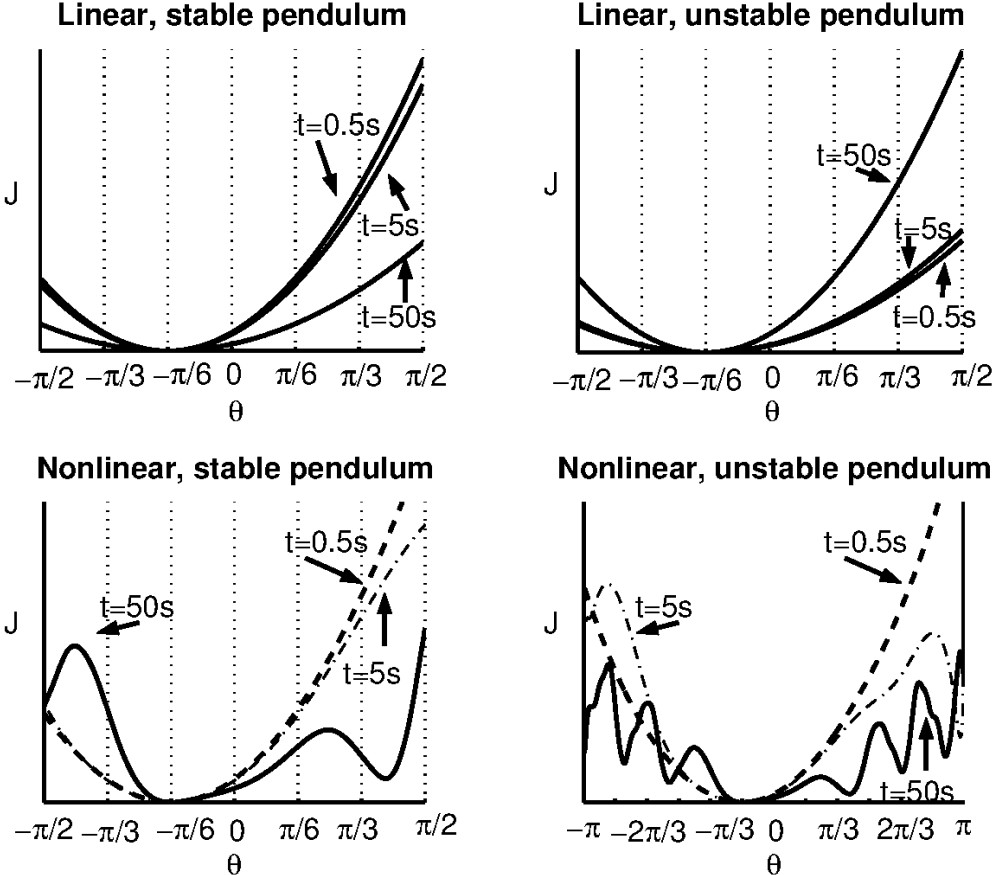

**Figure 8.** Cost function with respect to the initial pendulum angle. A synthetic observation was made from a model run with intial angle, $\theta = -\pi/6$. The time between the initial state and the cost function evaluation is 0.5, 5, or 50 seconds. *Upper left*: Linear, stable pendulum. *Upper right*: Linear, unstable pendulum *Lower left*: Nonlinear, stable pendulum. *Lower right*: Nonlinear, unstable pendulum. Notice the wider scale for $\theta$ in the lower, right panel. The pendulum's dynamical regimes are further explained in the text. Reproduced with permission from Gebbie (2004).





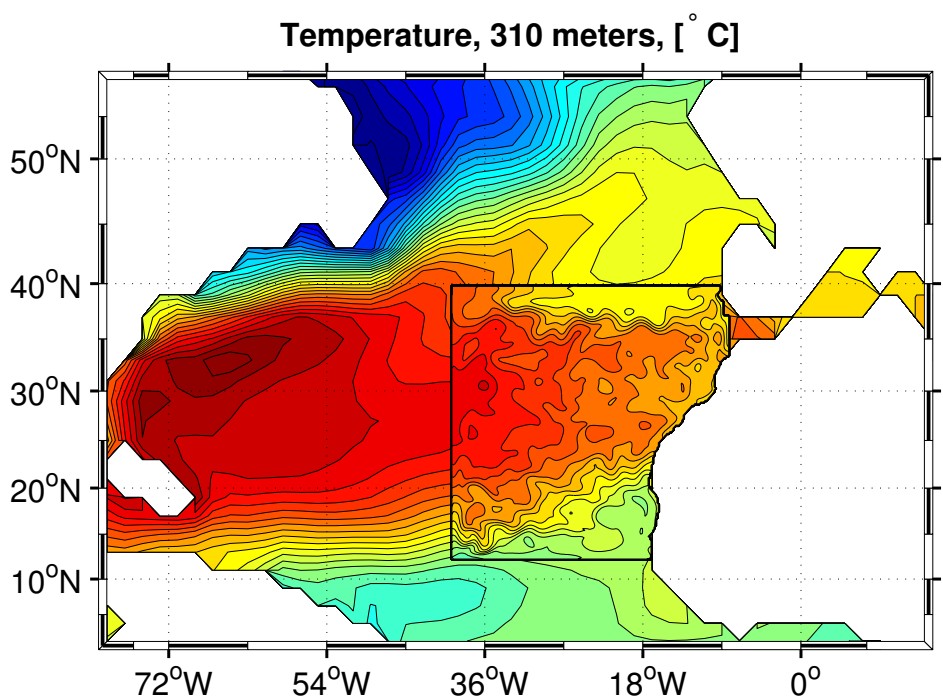

**Figure 9.** Nested view of the $1/6°$ regional state estimate of Gebbie et al. (2006) inside the $2°$ state estimate of Stammer et al. (2002b). Potential temperature at 310 meters depth, with a contour interval of $1°C$, is shown. The boundary between the two estimates (*thick black line*) is discontinuous in temperature because of the open-boundary control adjustments. Reproduced with permission from Gebbie (2004).