# Peer review of "Controllability, not chaos, key criterion for ocean state estimation"

_Nonlinear Processes in Geophysics, 2016_

## Referee Comment (RC1) · Anonymous Referee #1 · 26 Oct 2016

This manuscript contains a description of the results of a series of experiments in which 4DVAR was used to assimilate simulated data from a nonlinear system corresponding to the damped driven pendulum in a chaotic parameter regime. This example differs from most other examples in the literature of data assimilation in strongly nonlinear systems in that it is a non-autonomous system, unlike, say, Lorenz (63). With a few reservations, the example is fairly well worked out. The application of the $\chi^2$ test is particularly noteworthy. The basic results are worth publishing in some form.

The authors never state their model system explicitly. It is not (1). The system with which they are actually working differs from (1) in that it has a white noise term with variance $S_f$ (see (5)) added to the right hand side. The distinction is not trivial. I assume that the reference solution in their twin experiments is the stochastic system with the stochastic term set to zero. The effect of adding the unknown stochastic term

is to increase the number of degrees of freedom in the control space from two, i.e., the initial conditions in the purely deterministic problem, to the number of time steps taken by the numerical method, which is potentially infinite.

The general level of discussion in this manuscript might have been marginally acceptable twenty years ago, when implications of applications of techniques from the engineering world were still being explored, but most of the manuscript is far below the current state of the art.

Studies of chaotic systems forced by white noise have appeared in a number of places in the literature. One example can be found in a paper by Tziperman from the early 90s.

There is nothing novel about writing the 4DVAR cost function in terms of a Lagrange multiplier. The use of Lagrange multipliers in variational formulations of estimation and control problems has been in the engineering textbooks since the 70s, and appeared in the early work of Thacker in the ocean modeling literature. In the present context, in which the task is to estimate an unknown stochastic forcing function, the Lagrange multiplier formulation is valid, but the same Euler-Lagrange equations result from equivalent cost function formulations without Lagrange multipliers, see, e.g., the text by Kalnay or either of the books by Bennett, as well as many of the reviews in the literature.

The authors should note that the estimation problem is the dual of the control problem. General questions of linear controllability and observability are dealt with in engineering textbooks. This topic has been well worked out in the context of models of the ocean and atmosphere in the work of S. E. Cohn in the late 80s and early 90s. The question of nonlinear observability is very complex. There was a book by Casti on the subject published some time ago.

The question of dealing with underdetermination has been discussed extensively in the literature. Solutions to underdetermined problems are not, in general, unique. The

problem, in practice, is the fact that minimizing the cost function (5) involves searching a space of corrections that is potentially infinite. The highly irregular reconstructed forcing shown in the bottom panel of figure 4 is most likely one of an enormous number of minimizers of (5). There are almost certainly many others that will minimize the cost function, some smoother, many even more irregular.

Bennett showed that, in the linear problem, one solution can be found by choosing a correction to the first guess that lies in an $N_y$ dimensional space spanned by representer functions, where $N_y$ is the number of observations. This solution corresponds to the Moore-Penrose inverse. Arguments as to why that solution should be preferred over others are the stuff of textbooks.

Similar practical results can be had without explicit calculation of representers. In practical problems in modeling the ocean and atmosphere, the correction to the forcing function lies in a space of enormous dimension, so it is common to precondition the search for a cost function minimizer. This effectively reduces the dimension of the control space by choosing corrections to be a linear combination of singular vectors of the error covariance matrix. This approach is documented in the work of A. Lorenc and O. Talagrand. In the present problem, it might be reasonable to impose nontrivial temporal correlation on the forcing correction, which might have the effect of limiting the spectrum of the correction and thus ruling out irregular forcing corrections like that shown in figure 4.

The authors have a choice. They can simply report on the results of their twin experiments on their nonautonomous system and eliminate nearly all of the discussion, or they can go back over twenty or twenty five years of literature and rewrite the discussion to make it a meaningful contribution to the current state of the art.

---

## Referee Comment (RC2) · Anonymous Referee #2 · 3 Nov 2016

Main points: The perfect model and observations may not be sufficient in supporting the conclusions reached in this manuscript. Even though the reviewer agrees with the authors 4DVAR has its potential in oceanic state estimation, their case is simply too perfect for convincing the readers, particularly those from the non-variational analysis community, like EnKF or 4DEnVar. The reviewer suggests more realistic experiments and recommend this manuscript for major revision.

1. What are the values of ? From the true solution? I am afraid if ideal observations are used, it does not imply the conclusions made for this ideal model to be useful. 2. Page 4, line 23, The statement of "...the quantity inside curl brackets vanishes" is not generally true. To do so, there needs an additional term, penalizing the constraint. 3. Page 5, line 23. That is where the problem is that such as observation and prior knowledge and freely-running forward model are not enough. 4. For solving a global

minimization problem of (6), the first guess is crucial as the authors stated in page 4 line 25-26. However, the improved initial guess of their work presented in Section 2.4 cannot guarantee the initial guess is good enough for converging to the global minimizer. The authors should at least present convincing arguments of why they believe their improved initial guess could reach their goal. To the reviewer, the improved initial guess may fall into the same valley as the original initial guess.

---

## Referee Comment (RC3) · Anonymous Referee #3 · 16 Nov 2016

This paper investigates the key criterion for ocean state estimate, which is commonly called data assimilation (DA) in the oceanography and meteorological communities. There have been a lot of theoretical research and development over the past 30 years, the number of literature is just too many to list. The Lagrange multiplier method is just another way to express the minimization problem presented in the traditional 4D-Var. This can be found out Andrew Bennett's 1992 book: Inverse Methods in Physical Oceanography. Cambridge Monographs on Mechanics and Applied Mathematics. Cambridge University Press.

The reviewer does not agree the statement: "the dimensionality of many million state variables is not a a fundamental problem". I think both high dimensionality and nonlinearity of ocean models are challenging issues for the ocean prediction and data assimilation. The controllability in this article is very vaguely defined. In fact the importance

of boundary condition in ocean state estimation has long been recognized.

My view is that the system the authors have employed for the study and the 'observations' are too simplified to draw valuable conclusions for the DA research and development communities. The reviewer recommends a major revision to include more realistic models.

The presentation also needs improvement, there are some sentences that are either not correct or clear to the reader. e.g.

Page 2, line 34-35 is not correct.

Page 3, equation (1). It is stated that omega_d = 2/3, so the right hand side forcing f(t) is just a cosine function of time, I cannot see two independent variables (omega, theta) in this equation.

Page 5, lines 26-27: Kalman filter equation is normally solved by in lower space with the covariance represented by ensembles, there is no need for explicit representation.

Page 14, equations (A1), (A2). Omega is the time derivative of theta. They are not completely independent.

Page 25, the 2nd to the last sentence in Figure 7 caption is clear not a correct sentence.

---

## Author Comment (AC1) · 6 Jan 2017

**Reviewer #1**

- *This manuscript contains a description of the results of a series of experiments in which 4DVAR was used to assimilate simulated data from a nonlinear system corresponding to the damped driven pendulum in a chaotic parameter regime. This example differs from most other examples in the literature of data assimilation in strongly nonlinear systems in that it is a non-autonomous system, unlike, say, Lorenz (63). With a few reservations, the example is fairly well worked out. The application of the $\chi^2$ test is particularly noteworthy. The basic results are worth publishing in some form.*

We thank the reviewer for noting the care we took in this analysis, especially the $\chi^2$ posterior test, and for noting that the key results are worth publishing. We address some of your reservations in this point-by-point response.

- *The authors never state their model system explicitly. It is not (1). The system with which they are actually working differs from (1) in that it has a white noise term with variance $S_f$ (see (5)) added to the right hand side. The distinction is not trivial. I assume that the reference solution in their twin experiments is the stochastic system with the stochastic term set to zero. The effect of adding the unknown stochastic term is to increase the number of degrees of freedom in the control space from two, i.e., the initial conditions in the purely deterministic problem, to the number of time steps taken by the numerical method, which is potentially infinite.*

Thanks to the reviewer for spotting the error in defining the external forcing term. The equation is deterministic and is now stated explicitly in equation (1). The weight matrix $S_f$ in equation (5) is selected in the state estimation process to limit the difference of the improved guess from the first guess, similar to the formulation in Bennett (2002). The revised text reads as follows.

> The motion of the forced pendulum is described by the deterministic equation (*Baker and Gollub*, 1990),
>
> $$\frac{d^2\theta}{dt^2} + \frac{1}{q}\frac{d\theta}{dt} + \frac{g}{l}\sin\theta = f(t), \tag{1}$$
>
> where $\theta$ is the displacement angle from vertical, $q$ is a damping coefficient, $g$ is gravitational acceleration, $l$ is the pendulum length, and $f(t)$ is an external forcing term. In turn, the external forcing has a first guess and a perturbation, $f(t) = f_0 + \delta f(t)$, where the first-guess is set to periodic forcing, $f_0(t) = b\cos(\omega_d t)$.

- *The general level of discussion in this manuscript might have been marginally acceptable twenty years ago, when implications of applications of techniques from the engineering world were still being explored, but most of the manuscript is far below the current state of the art.*

It is clear that the motivation and goals of the manuscript need to be made more explicit. The Lagrange multiplier method is popular in oceanography due to automatic adjoint model compilers and strategies to reduce computer memory consumption. Much time and effort has been spent to develop this technique in real-world scenarios, yet it is unclear whether this method should be applied to eddy-resolving models and how long the time window should be. For the Lagrange multiplier method to be successful in state-of-the-art ocean models, two major issues need to be addressed: (1) the high dimensionality of the forward model and estimation problem, and (2) the nonlinearity of ocean models at increasingly fine resolution. Issue (1) has been overcome by groups such as the ECCO Consortium. Here we focus on (2). It is true that issues may arise by the combined effect of (1) and (2), but first we attempt to isolate the effect of nonlinearity.

With this problem in mind, it is logical to find a numerical model that can be thoroughly understood and one that is highly nonlinear. It is not the goal of this manuscript to use a state-of-the-art numerical model. We believe that these expectations should be set at the outset, so we include the following in the Introduction.

Because the effect of nonlinearity is seen as the major roadblock for application of the Lagrange multipler method, we isolate this effect by choosing a model that is highly nonlinear but low-dimensional: the forced, chaotic pendulum (Section 2). Toy models are worth revisiting because the dynamics are comparatively simple to understand, and they have strongly influenced when the Lagrange multiplier method has been deployed to realistic ocean problems. We will show that previous toy models have sometimes been misinterpreted.

We now also emphasize upfront that the development of a new state-of-the-art data assimilation technique is not the goal of this work, either. Instead, we wish to evaluate the current use of the Lagrange multiplier method. Now, the Introduction makes this explicit.

Rather than developing a new state-of-the-art data assimilation technique, we proceed by taking the existing Lagrange multipler method and developing diagnostics regarding when and why it succeeds or fails, as evaluated by the ability to fit observations. Relative to the initialization problem, the prospects for a successful state estimate are shown to be improved in the boundary control problem, even if one uses a highly nonlinear model such as the forced, chaotic pendulum (Section 3).

The typical criterion for successful state estimation has been the stability to initial perturbation. In the manuscript, we provide a counterexample showing that state estimation can be successful for an unstable system when it is controllable. This is one novel result we are reporting, and the previous works suggested by the reviewers have not already made this point, nor do they appear to contradict it.

- *Studies of chaotic systems forced by white noise have appeared in a number of places in the literature. One example can be found in a paper by Tziperman from the early 90s.*

We now mention Tziperman's work on chaotic systems as a motivating factor in using our toy model.

Toy models are worth revisiting because the dynamics are comparatively simple to understand, the nonlinear coupling to periodic forcing has been shown to be important in atmosphere-ocean dynamics (e.g., *Tziperman et al.*, 1994), and these models have strongly influenced when the Lagrange multiplier method has been deployed to realistic ocean problems.

- *There is nothing novel about writing the 4DVAR cost function in terms of a Lagrange multiplier. The use of Lagrange multipliers in variational formulations of estimation and control problems has been in the engineering textbooks since the 70s, and ap- peared in the early work of Thacker in the ocean modeling literature. In the present context, in which the task is to estimate an unknown stochastic forcing function, the Lagrange multiplier formulation is valid, but the same Euler-Lagrange equations result from equivalent cost function formulations without Lagrange multipliers, see, e.g., the text by Kalnay or either of the books by Bennett, as well as many of the reviews in the literature.*

The reviewer's point that the Lagrange multiplier method is just another way to express a minimization problem is in line with the point of our Section 4.1. In that section, we suggest that the criteria for successful use of the Kalman Filter/Smoother, which also minimizes the same cost function, is the same as that for the Lagrange multiplier method. In particular, Sec. 4.1 states the following.

Our results suggest that the equivalence of the Kalman filter/smoother and Lagrange multiplier method may be extended to nonlinear problems, thus explaining why the chaotic estimation problem may be solved by the Lagrange multiplier method.

- *The authors should note that the estimation problem is the dual of the control problem. General questions of linear controllability and observability are dealt with in engineering textbooks. This topic has been well worked out in the context of models of the ocean and atmosphere in the work of S. E. Cohn in the late 80s and early 90s. The question of nonlinear observability is very complex. There was a book by Casti on the subject published some time ago.*

We thank the reviewer for the excellent suggestions for further references. In addition, we now point out the duality of the estimation and the control problem. As pointed out by the reviewer, observability and controllability conditions for nonlinear state estimation are difficult problems. Results for certain nonlinear systems are found in the book by *Casti* (1985) (which evolves from the review paper of *Casti* (1982)). For general nonlinear estimation problems, the down-gradient-based iterative optimization is likely one of the best methods, and we have shown the relevance of controllability to the iterative process in our example. We include the following new paragraph in the Discussion.

> To recover the true trajectory of a system, observability is also important, as the estimation problem is the dual of the control problem (*Fukumori et al.*, 1993; *Marchal*, 2014). For the linear problem, *Cohn and Dee* (1988) showed that completely observability implies asymptotic stability of the Kalman filter/smoother. Defining observability and controllability conditions for nonlinear state estimation problems is difficult *Casti* (1985). In practice, the important criterion is ability to solve equation (10). Strictly speaking, the solution criteria will therefore depend upon both the controllability matrix, **C**, and the observational matrix, **E**, which combines the issues of observability and controllability. Here, we suggest the operational definition that a system is effectively controllable when the solution to (10), generalized to multiple observations, exists.

- *The question of dealing with underdetermination has been discussed extensively in the literature. Solutions to underdetermined problems are not, in general, unique. The problem, in practice, is the fact that minimizing the cost function (5) involves searching a space of corrections that is potentially infinite. The highly irregular reconstructed forcing shown in the bottom panel of figure 4 is most likely one of an enormous number of minimizers of (5). There are almost certainly many others that will minimize the cost function, some smoother, many even more irregular.*

The reviewer's point about the underdetermined nature of the problem is consistent with our discussion in Sec. 3.3, where we acknowledge that the solution is not unique, but we focus on finding any acceptable fit. We view the problem as having two clear steps. It is a first step to find any solution that acceptably fits the data. Only then can we proceed to investigate the uniqueness of the solution. In real-world situations, the first step may be the only one that is practical.

- *Bennett showed that, in the linear problem, one solution can be found by choosing a correction to the first guess that lies in an N y dimensional space spanned by repre- senter functions, where N y is the number of observations. This solution corresponds to the Moore-Penrose inverse. Arguments as to why that solution should be preferred over others are the stuff of textbooks.*

It is worth clarifying that the external forcing had been treated as a controllable parameter in many works in the literature. In the Introduction, we now state the following.

> As has been documented in detail by many authors including the textbook of *Bennett* (1992), the ocean state estimation problem is better described as a time-variable boundary value problem because synoptic atmospheric variability acts as an external forcing on the ocean (Section 2). Given our relatively uncertain knowledge regarding air-sea fluxes, the ocean state estimation is rightfully considered a time-variable boundary value problem where both the initial conditions and boundary conditions must be found. For example, *Bennett* (2002) described an estimation method for the external forcing, initial and boundary conditions that solves the Euler-Lagrange equations for a linear model.

- *Similar practical results can be had without explicit calculation of representers. In practical problems in modeling the ocean and atmosphere, the correction to the forcing function lies in a space of enormous dimension, so it is common to precondition the search for a cost function minimizer. This effectively reduces the dimension of the con- trol space by choosing corrections to be a linear combination of singular vectors of the error covariance matrix. This approach is documented in the work*

*of A. Lorenc and O. Talagrand. In the present problem, it might be reasonable to impose nontrivial temporal correlation on the forcing correction, which might have the effect of limiting the spec- trum of the correction and thus ruling out irregular forcing corrections like that shown in figure 4.*

As the reviewer suggests, temporal correlations in the forcing field can be imposed through the use of nondiagonal weighting matrix, $\mathbf{S}_f$, in the cost function. The revised manuscript now describes how temporal correlations have been enforced in our analysis. The following material has been added to Sec. 3.4.

> We investigate the effect of a decrease in the number of controls by redefining the external forcing control perturbation. For $N_u$ forcing controls, we define,
>
> $$f(t) = f_0(t) + \Gamma(t) \begin{pmatrix} \delta f(0) \\ \delta f(T/N_u) \\ \delta f(2T/N_u) \\ \vdots \\ \delta f(T) \end{pmatrix}, \tag{2}$$
>
> where $\Gamma(t)$ is a matrix that performs linear interpolation in time, and $\delta f(t)$ is only defined at $N_u$ control times. This formulation enforces some temporal correlation in the external forcing. Alternatively, this could be accomplished using a nondiagonal weighting matrix, $\mathbf{S}_f$.

- *The authors have a choice. They can simply report on the results of their twin exper- iments on their nonautonomous system and eliminate nearly all of the discussion, or they can go back over twenty or twenty five years of literature and rewrite the discussion to make it a meaningful contribution to the current state of the art.*

We have taken seriously the reviewer's suggestions to place our work in the greater context of the published literature. Major revisions include new paragraphs in the Introduction regarding the motivation and aims, more information about the background of controllability that places our work in a broader context, and a new figure using an imperfect model that recreates a more realistic scenario.

---

## Author Comment (AC2) · 6 Jan 2017

**Reviewer #2**

- *Main points: The perfect model and observations may not be sufficient in supporting the conclusions reached in this manuscript. Even though the reviewer agrees with the authors 4DVAR has its potential in oceanic state estimation, their case is simply too perfect for convincing the readers, particularly those from the non-variational analysis community, like EnKF or 4DEnVar. The reviewer suggests more realistic experiments and recommend this manuscript for major revision.*

Thanks to the reviewer for the interesting and helpful perspective. The experimental setup may not be as idealized as originally thought. For example, the synthetic observations generated in this study are not perfect, and we detail their generation in the response to the next point below.

Regarding the perfect model assumption, our equations have been formulated in analogy to the ECCO (Estimating the Climate and Circulation of the Ocean) state estimation equations (*Stammer et al.*, 2002), as our goal is to develop diagnostics regarding when and why the Lagrange multiplier method succeeds or fails. In the ECCO formulation, the ocean model equations are typically treated as perfect in the ocean interior, and errors are permitted in the surface gridcells with air-sea forcing. Here, we permit these errors and attribute them to errors in the external forcing.

One aspect of the analysis that we have improved is the first-guess of the forcing field. In a case study where the first-guess of the forcing is zero, the results are similar to the original case. This is reported in a new Section 3.5 and a new Figure 8. The new section follows.

> The previous examples in Section 3 proceed with prior information that the forcing is periodic with an accurate magnitude and phase. A good analogy is the regular forcing of solar insolation on the ocean surface. Here, we test the performance of the Lagrange multiplier method with inaccurate prior information about the forcing, as is a more realistic analogy to the uncertainty of air-sea fluxes. In particular, our first guess of the forcing, $f_0(t)$, is systematically biased by decreasing $b$ from 1.5 to 0.75 rad s$^{-2}$. The trajectory driven by inaccurate forcing is no worse than the previous cases with accurate forcing due to the dominance of the chaotic dynamics of system (Figure 8). Using the same observations as shown in Figure 3, we find that the chaotic pendulum trajectory is tracked over multiple nonlinear timescales despite this more stringent test. In this case, however, the forcing estimate still contains errors relative to the true forcing calculated with $b = 1.5$ rad s$^2$, and some high-frequency structures remain in $f(t)$ (see "improved first guess" in bottom panel, Figure 8). If instead the Lagrange multiplier method is started from the standard first guess, a smoother and more accurate estimate of the forcing is obtained at the expense of not fitting the data as well (see "final estimate" in bottom panel). Any remaining irregular structures can be handled by imposing temporal correlations as was done in Section 3.4. If such measures are not taken, the investigator must take care to decide what elements of the forcing represent true variability and which are compensating for model error. In our simple system of equations, model errors and forcing errors are mathematically equivalent. In state estimates with eddy-resolving GCMs, however, smallscale forcing variability is found near oceanic fronts and the investigator must determine on a case-by-case basis to what extent it reflects real variability.

Convincing the non-variational analysis community to adopt the Lagrange multiplier method would be a challenging task, but outside the goals of our work. It is clear that the motivation and goals of the manuscript need to be made more explicit. The Lagrange multiplier method is popular in oceanography due to automatic adjoint model compilers and strategies to reduce computer memory consumption. Much time and effort has been spent to develop this technique in real-world scenarios, yet it is unclear whether this method should be applied to eddy-resolving models and how long the time window should be. For the Lagrange multiplier method to be successful in state-of-the-art ocean models, two major issues need to be addressed: (1) the high dimensionality of the forward model and estimation problem, and (2) the nonlinearity of ocean models at increasingly fine resolution. Issue

(1) has been overcome by groups such as the ECCO Consortium. Here we focus on (2). It is true that issues may arise by the combined effect of (1) and (2), but first we attempt to isolate the effect of nonlinearity.

With this problem in mind, it is logical to find a numerical model that can be thoroughly understood and one that is highly nonlinear. It is not the goal of this manuscript to use a state-of-the-art numerical model. We believe that these expectations should be set at the outset, so we include the following in the Introduction.

> Because the effect of nonlinearity is seen as the major roadblock for application of the Lagrange multipler method, we isolate this effect by choosing a model that is highly nonlinear but low-dimensional: the forced, chaotic pendulum (Section 2). Toy models are worth revisiting because the dynamics are comparatively simple to understand, and they have strongly influenced when the Lagrange multiplier method has been deployed to realistic ocean problems. We will show that previous toy models have sometimes been misinterpreted.

We now also emphasize upfront that the development of a new state-of-the-art data assimilation technique is not the goal of this work, either. Instead, we wish to evaluate the current use of the Lagrange multiplier method. Now, the Introduction makes this explicit.

> Rather than developing a new state-of-the-art data assimilation technique, we proceed by taking the existing Lagrange multipler method and developing diagnostics regarding when and why it succeeds or fails, as evaluated by the ability to fit observations. Relative to the initialization problem, the prospects for a successful state estimate are shown to be improved in the boundary control problem, even if one uses a highly nonlinear model such as the forced, chaotic pendulum (Section 3).

Thus, a practical goal of this work is to convince those groups that already use the Lagrange multiplier method to reconsider the range of scenarios in which they apply the method.

- *1. What are the values of ? From the true solution? I am afraid if ideal observations are used, it does not imply the conclusions made for this ideal model to be useful.*

Stochastic noise is used to generate synthetic data, mimicking the imperfection of ocean observations. This is a common approach that has appeared in ocean state estimation studies such as Tziperman et al. (1992), who used the same iterative adjoint method on a simplified ocean GCM (the momentum equations are balanced and the nonlinear advection is neglected). The manuscript states the following.

> We consider an "identical twin" experiment where the true solution is known (solid line, Figure 1), and we observe the pendulum angle episodically through time with normally-distributed random errors of standard deviation, $\sigma_\theta = 0.5$ rad. In most oceanographically-relevant cases, observations have already been collected over some fixed time interval ($0 \leq t \leq T$). Here, observations, $y(t)$, are taken at a set of $N_y$ evenly-spaced times with an time interval of $\Delta t_y = T/(N_y - 1)$.

- *2. Page 4, line 23, The statement of . . .the quantity inside curl brackets vanishes is not generally true. To do so, there needs an additional term, penalizing the constraint.*

The terms in the curly bracket vanish by definition of our time-stepping model in equation (2). There was an error in defining the external forcing term which may have caused confusion. The equation is deterministic and is now stated explicitly in equation (1). The weight matrix $S_f$ in equation (5) is selected in the state estimation process to limit the difference of the improved guess from the first guess, similar to the formulation in Bennett (2002). The revised text reads as follows.

> The motion of the forced pendulum is described by the deterministic equation (*Baker and*

*Gollub*, 1990),

$$\frac{d^2\theta}{dt^2} + \frac{1}{q}\frac{d\theta}{dt} + \frac{g}{l}\sin\theta = f(t), \qquad (3)$$

where $\theta$ is the displacement angle from vertical, $q$ is a damping coefficient, $g$ is gravitational acceleration, $l$ is the pendulum length, and $f(t)$ is an external forcing term. In turn, the external forcing has a first guess and a perturbation, $f(t) = f_0 + \delta f(t)$, where the first-guess is set to periodic forcing, $f_0(t) = b\cos(\omega_d t)$.

- *3. Page 5, line 23. That is where the problem is that such as observation and prior knowledge and freely-running forward model are not enough.*

We agree with the reviewer that the first-guess may not always be sufficient to track a chaotic system. For this reason, we implement a $\chi - 2$ test, detailed in the next point-by-point response, that diagnoses the likelihood of success or failure.

- *4. For solving a global minimization problem of (6), the first guess is crucial as the authors stated in page 4 line 25-26. However, the improved initial guess of their work presented in Section 2.4 cannot guarantee the initial guess is good enough for converging to the global minimizer. The authors should at least present convincing arguments of why they believe their improved initial guess could reach their goal. To the reviewer, the improved initial guess may fall into the same valley as the original initial guess.*

The reviewer correctly states that there is no guarantee that a solution will be a global minimizer. We discuss the underdetermined nature of the problem in Sec. 3.3, where we acknowledge that the solution is not unique. Instead, we focus on finding any acceptable fit. We view the problem as having two clear steps. It is a first step to find any solution that acceptably fits the data. Only then can we proceed to investigate the uniqueness of the solution. In real-world situations, the first step may be the only one that can actually be evaluated.

To consider whether a solution is an acceptable fit, we include Figure 6 which details the size of the cost function for various numbers of controls and observations. By implementing a $\chi^2$ posterior statistical test, we determine the ratio of success to failure for various parameter ranges. After running many trials, we do not guarantee the results for any particular number of observations and controls, but a clear pattern emerges. We suggest that the pattern of Figure 6 is explained by the basic metrics of controllability and observability. rather than the stability of the system. This is one novel result we are reporting, and the previous works suggested by the reviewers have not already made this point, nor do they appear to contradict it.

---

## Author Comment (AC3) · 6 Jan 2017

**Reviewer #3**

- *This paper investigates the key criterion for ocean state estimate, which is commonly called data assimilation (DA) in the oceanography and meteorological communities. There have been a lot of theoretical research and development over the past 30 years, the number of literature is just too many to list. The Lagrange multiplier method is just another way to express the minimization problem presented in the traditional 4D- Var. This can be found out Andrew Bennetts 1992 book: Inverse Methods in Physi- cal Oceanography. Cambridge Monographs on Mechanics and Applied Mathematics. Cambridge University Press.*

The reviewer is clearly correct in their assessment of the relationship between 4D-VAR and the Lagrange multiplier method. We prefer the notation, "Lagrange multiplier method," because it will hopefully be understood by scientists in the greater mathematics and physics communities. We include a sentence in the Introduction that shows that these terms are interchangeable.

> The Lagrange multiplier method (e.g., *Thacker and Long*, 1988; *Wunsch*, 2010), sometimes called the adjoint method (e.g., *Hall et al.*, 1982; *Tziperman and Thacker*, 1989), "4D-VAR" (e.g., *Courtier et al.*, 1994; *Ferron and Marotzke*, 2003), or variational data assimilation (e.g., *LeDimet and Talagrand*, 1986; *Bonekamp et al.*, 2001; *Bennett*, 2002), is a method that satisfies both of these criteria, unlike the Kalman filter (e.g., *Fukumori and Malanotte-Rizzoli*, 1995) or nudging techniques (e.g., *Malanotte-Rizzoli and Tziperman*, 1995).

The reviewer's point that the Lagrange multiplier method is just another way to express a minimization problem is in line with the point of our Section 4.1. In that section, we suggest that the criteria for successful use of the Kalman Filter/Smoother, which also minimizes the same cost function, is the same as that for the Lagrange multiplier method.

- *The reviewer does not agree the statement: the dimensionality of many million state variables is not a a fundamental problem. I think both high dimensionality and nonlinearity of ocean models are challenging issues for the ocean prediction and data assimilation. The controllability in this article is very vaguely defined. In fact the importance of boundary condition in ocean state estimation has long been recognized.*

Some of the sentences in the Introduction may have given the impression that we are the first to consider the ocean state estimation problem as a time-variable boundary control problem. Given our long list of references on the topic, this is obviously not true! We hope to strike a better balance by revising the final paragraph of the Introduction as follows.

> As has been documented in detail by many authors including the textbook of *Bennett* (1992), the ocean state estimation problem is better described as a time-variable boundary value problem because synoptic atmospheric variability acts as an external forcing on the ocean (Section 2).

The Estimating the Climate and Circulation of the Ocean (ECCO) Consortirum has specifically focused on large-scale ocean state estimation with ocean models of coarse-enough resolution that they are essentially linear (e.g., *Stammer*, 1997; *Stammer et al.*, 2002). In this context, *Wunsch and Heimbach* (2007) stated, "The main issue for the oceanographic problem is one of dimension," in accordance with the reviewer's comment. The success of the ECCO Consortium, insofar as a solution can be found that fits the ocean data, indicates that the dimensionality of the problem can be overcome and therefore is not a fundamental obstacle. Here we revise the sentence.

> Research conducted by the ECCO (Estimating the Circulation and Climate of the Ocean) Consortium (*Stammer et al.*, 2002, 2004) has demonstrated that (1), the dimensionality of many million state variables presents a challenge, but, insofar as a solution can be found that fits the ocean data, it can be overcome and it is does not pose a fundamental obstacle.

Observability and controllability conditions for nonlinear state estimation are difficult problems. Results for certain nonlinear systems are found in the book by *Casti* (1985) (which evolves from the review paper of *Casti* (1982)). For general nonlinear estimation problems, the down-gradient-based iterative optimization is likely one of the best methods, and we have shown the relevance of controllability to the iterative process in our example. We include the following new paragraph in the Discussion that also expands upon observability and controllability.

> To recover the true trajectory of a system, observability is also important, as the estimation problem is the dual of the control problem (*Fukumori et al.*, 1993; *Marchal*, 2014). For the linear problem, *Cohn and Dee* (1988) showed that completely observability implies asymptotic stability of the Kalman filter/smoother. Defining observability and controllability conditions for nonlinear state estimation problems is difficult *Casti* (1985). In practice, the important criterion is ability to solve equation (10). Strictly speaking, the solution criteria will therefore depend upon both the controllability matrix, **C**, and the observational matrix, **E**, which combines the issues of observability and controllability. Here, we suggest the operational definition that a system is effectively controllable when the solution to (10), generalized to multiple observations, exists.

- *My view is that the system the authors have employed for the study and the observations are too simplified to draw valuable conclusions for the DA research and development communities. The reviewer recommends a major revision to include more realistic models.*

Our experimental setup may not be as simple as the reviewer understood. We note that stochastic noise is used to generate the synthetic observations, mimicking the imperfect nature of ocean observations. This is a common approach that has appeared in ocean state estimation studies such as Tziperman et al. (1992).

Regarding the simplified model of this study, it has been explicitly chosen with the motivation and goals of the manuscript in mind. Much time and effort has been spent to develop the Lagrange multiplier method in real-world scenarios, yet it is unclear whether this method should be applied to eddy-resolving models and how long the time window should be. For the Lagrange multiplier method to be successful in state-of-the-art ocean models, two major issues need to be addressed: (1) the high dimensionality of the forward model and estimation problem, and (2) the nonlinearity of ocean models at increasingly fine resolution. Issue (1) has been overcome by groups such as the ECCO Consortium, leading us to focus on (2).

With this problem in mind, it is logical to find a numerical model that can be thoroughly understood and one that is highly nonlinear. It is not the goal of this manuscript to use a state-of-the-art numerical model. We believe that these expectations should be set at the outset, so we include the following in the Introduction.

> Because the effect of nonlinearity is seen as the major roadblock for application of the Lagrange multipler method, we isolate this effect by choosing a model that is highly nonlinear but low-dimensional: the forced, chaotic pendulum (Section 2). Toy models are worth revisiting because the dynamics are comparatively simple to understand, and they have strongly influenced when the Lagrange multiplier method has been deployed to realistic ocean problems. We will show that previous toy models have sometimes been misinterpreted.

We now also emphasize upfront that the development of a new state-of-the-art data assimilation technique is not the goal of this work, either. Instead, we wish to evaluate the current use of the Lagrange multiplier method. Now, the Introduction makes this explicit.

> Rather than developing a new state-of-the-art data assimilation technique, we proceed by taking the existing Lagrange multipler method and developing diagnostics regarding when

and why it succeeds or fails, as evaluated by the ability to fit observations. Relative to the initialization problem, the prospects for a successful state estimate are shown to be improved in the boundary control problem, even if one uses a highly nonlinear model such as the forced, chaotic pendulum (Section 3).

The typical criterion for successful state estimation has been the stability to initial perturbation. In the manuscript, we provide a counterexample showing that state estimation can be successful for an unstable system when it is controllable. This is one novel result we are reporting, and the previous works suggested by the reviewers have not already made this point, nor do they appear to contradict it.

One aspect of the analysis that we have improved is the first-guess of the forcing field. In a case study where the first-guess of the forcing is zero, the results are similar to the original case. This is reported in a new Section 3.5 and a new Figure 8.

Without doubt, it is a worthy goal to demonstrate the importance of controllability for the Lagrange multiplier method with a more complex geophysical model. Based on the results of this manuscript, a thorough test with that type of model is a logical next step, as suggested in Sec. 5.

- *The presentation also needs improvement, there are some sentences that are either not correct or clear to the reader. e.g. Page 2, line 34-35 is not correct.*

We believe lines 34-35 to be grammatically correct as originally formulated.

- *Page 3, equation (1). It is stated that $\omega_d = 2/3$, so the right hand side forcing $f(t)$ is just a cosine function of time, I cannot see two independent variables ($\omega$, $\theta$) in this equation.*

Certainly the angular velocity and displacement are related by a time derivative and are not independent. Those two variables describe the state of the model, and their time tendencies depend on one another, constituting an algebraic system of equations (discrete version of a system of differential equations). We find no mention of independence in the text and we do not believe that the dependence impacts the relevance of this model to state estimation.

- *Page 5, lines 26-27: Kalman filter equation is normally solved by in lower space with the covariance represented by ensembles, there is no need for explicit representation.*

The reviewer makes a nice point that we have now included in the manuscript:

> One remedy is to solve the Kalman filter equation in reduced space with the covariance represented by ensembles rather than being explicitly represented. Instead, we design a whole-domain method that is computationally efficient and provides a good first guess for the boundary control problem.

- *Page 14, equations (A1), (A2). Omega is the time derivative of theta. They are not completely independent.*

We agree with the reviewer's statement, but admit that we are unclear as to the larger relevance of this statement.

- *Page 25, the 2nd to the last sentence in Figure 7 caption is clear not a correct sentence.*

We have revised the sentence to the following.

> The local minimum in the first two dimensions is no longer an extremum in the other two dimensions or the combined three-dimensional space.

---

## Author Response (AR2)

WOODS HOLE OCEANOGRAPHIC INSTITUTION
DEPARTMENT OF PHYSICAL OCEANOGRAPHY
MS #29, WOODS HOLE, MASSACHUSETTS, 02543
TEL: (508) 289-2801, ggebbie@whoi.edu
Geoffrey Gebbie

April 28, 2017

*Nonlinear Processes in Geophysics*
European Geophysical Union

Dear Dr. Zoltan Toth, Editor of *Nonlinear Processes in Geophysics*,

Please find under this cover a revision of the manuscript entitled, "Controllability, not chaos, key criterion for ocean state estimation," by Geoffrey Gebbie and Tsung-Lin Hsieh. We have found the exchange with you and the two reviewers to be stimulating and thought-provoking. The constructive and clarifying comments that we have received have undoubtedly improved the manuscript, and for that, my co-author Tsung-Lin and I are grateful.

In particular, we have worked to make the manuscript more accessible to the general readership of *NPG*. We have added extra comments regarding some of the more unusual technical aspects of the manuscript, including the method of total inversion. We have also added clear language about the aims and goals of the work: to find any solution that reasonably fits the data. Of course, one could easily define further goals that would extend our work.

We are grateful to the detailed level of technical comments provided by Reviewer 2. In particular, we have remade Figure 6 from scratch, taking into account the differences between $J$ and $J_d$. After accounting for these reviews, we believe that our $\chi^2$ statistical test is more rigorous and clear to the general community.

We have also responded to Reviewer #1's comments, and agree that this work should be repeated with a more sophisticated geophysical model. We believe that these comments from Reviewer #1 actually show that our work has been successful, as such an avenue of research was not considered viable before we performed our experiments.

Thank you again for guiding this work through the peer-review process. We appreciate the opportunity to improve, clarify, and defend our work. A detailed point-by-point response to the reviewers follows.

Sincerely,

Geoffrey (Jake) Gebbie,
on behalf of Tsung-Lin Hsieh

**Editor's comments, February 2017**

- *p2 l8 - "and it does not pose"*

  This has been stricken from the manuscript in response to Report #2 and the discussion of whether the obstacle is fundamental.

- *p9, l35: "The nonlinear timescale here IS defined"*

  Done.

- *p12, l30 - "COMPLETE observability"*

  Done.

- *p13 l15 - "In THIS section"*
  Done.

- *Regarding Section 3.3 and the underdetermined nature of the problem: While I agree the Rev's point is consistent with your discussion in Section 3.3, perhaps you could use more direct language in Section 3.3 (or elsewhere, similar to your response to the Reviewer's comment?)*

  We think that the editor's suggestion is important enough that it should be stated as upfront as possible. Thus, we add the following sentences to Section 2.5.

  > Forward-adjoint model integrations are repeated until $J$ has an acceptable value by a $\chi^2$ statistical test. Here we consider the state estimation method successful if any solution that acceptably fits the data is found. We acknowledge that many of the cases presented here are underdetermined, and thus we expect those solutions to not be unique. We emphasize, however, that finding any solution would be a breakthrough, as this test has been difficult to satisfy with chaotic models and the Lagrange multiplier method.

- *Perhaps you could clarify, as you define "controllability" on p.3, that in your study, you control the system via adjustments to the atmospheric boundary forcing?*

  We have implemented this nice suggestion by adding a new sentence to the Introduction.

  > The control variables of the pendulum are analogous to adjustments of the atmospheric boundary forcing in an ocean model.

- *comments on p3 & p14 - perhaps in passing by, the dependent relationship between omega and theta could be pointed out in the manuscript*

  We have added a clarifying phrase to the following sentence on Page 3.

  > Following the numerical implementation in Appendix A, the state vector is defined, $\mathbf{x}(t) = [\omega(t)\ \theta(t)]^T$, where $^T$ is the vector transpose and the state variables are related by, $\omega = d\theta/dt$.

**Report #1, Reviewer #3**

- *It is too hard for the reviewer to accept the authors statement that the high dimensionality of the operational model is overcome and, . is not a fundamental obstacle. In my opinion, high dimensionality of realistic ocean and atmospheric models is one of the fundamental challenges for data assimilation, as the DA system struggles to fit the high-dimension model states to the irregularly distributed observations during the minimization process.*

We respect the reviewer's opinion that the challenges posed by the dimensionality of geophysical problems should not be minimized. We see the point that the dimensionality could still be considered a "fundamental" issue. To clarify our writing, we strike this reference to a "fundamental obstacle", and provide additional context to our statement by referencing a particular discussion from the literature that supports our point of view. The new sentence follows.

> Research conducted by the ECCO (Estimating the Circulation and Climate of the Ocean) Consortium (*Stammer et al.*, 2002, 2004) has demonstrated that (1), the dimensionality of many million state variables presents a challenge, but it can be overcome insofar as a solution can be found that fits the ocean data (e.g., *Wunsch and Heimbach*, 2007).

The first sentence of the Conclusions is also updated to reflect a more-narrow, more-precise interpretation of our findings.

> Nonlinearity is not a fundamental obstacle to constraining a model to observations using the Lagrange multiplier method.

In the big picture, we introduce two major types of issues with nonlinear geophysical state estimation. We provide some insight to our thought process in the Introduction, namely that issue 2 (i.e., nonlinearity) is a priority and that we seek to address it. Should a reader disagree and feel that issue 1 (i.e., dimensionality) is also important, it does not eliminate the need to address issue 2, as we have done in this manuscript.

**Report #2, Reviewer #1**

- *In the revised version the authors have answered many of my initial questions, but some questions remain and the revisions have raised new questions. As I now understand it, the major point of the paper is that earlier results that seemed to show that 4DVAR didnt work so well in chaotic systems is that, in the cases cited, the control space didnt have enough degrees of freedom. With insufficient degrees of freedom in the control space, the problem becomes that of finding a minimum in the surface depicted in figure 2. From this point of view, the punch line to the manuscript is figure 7.*

We thank the reviewer for the concise summary of our major findings. We agree that Figure 2 illustrates the issue with the initialization problem. We consider the main punchline, however, to be Figure 6, not 7. Figure 6 synthesizes many different cases and shows the clear dependence of the observational fit on the number of controls. By its construction with many different experiments, we believe it is robust. Figure 7 is an example case, where we have found that a local minimum vanishes due to the increase in control-space dimension. As it is very difficult to manually determine the prevalence of this effect, we do not want to overstate its importance in the Abstract and Introduction. As the Abstract is now written, it properly highlights Figure 6, which we believe to be the true punchline.

- *The conclusion that initial conditions are not sufficient to control a chaotic system in many cases of interest is well known, but its useful to see it stated succinctly in one place. The philosophical problem I have with the approach taken in this manuscript is that it seems to imply that a control space of greater dimension is always preferable to one of lesser dimension. This is definitely not the case. For the linear problem, the dimension of the control space should not exceed the number of observations, and conditioning issues will usually restrict the dimension further. This is not necessarily so for non-linear problems, but it is a useful guide.*

We agree with the reviewer that the choice of the number of control variables is largely philosophical. In the typical ocean state estimation problem, the number of controls depends upon the spatial and temporal dimensions of the air-sea fluxes. The independent number of controls, or the degrees of freedom in the air-sea fluxes, may be reduced due to spatial and temporal correlations, as the reviewer has noted elsewhere. In a realistic problem, we believe that these dimensions are typically very large, as small-scale perturbations may affect the large-scale circulation or otherwise sensitively impact the observational sites.

Although this discussion may be peripheral to the manuscript, we feel that it might be useful to describe our personal philosophy on the number of control variables. Consider the case where "too many" control variables are permitted. In this case, the observations do not provide any information about a subset of the controls. While this seems to be a poor result, the updated control uncertainty will not be much reduced from the prior errors put on the controls. As long as the investigator takes this error calculation into consideration, there is no worry that the control estimate will be overinterpreted. In this conceptual case where the error statistics are handled accurately, we don't see why the number of controls should be restricted to the number of observations. (Of course it is no small feat to get the error statistics correct!)

We have also experienced situations where the restriction of the number of control variables ends up misleading the investigator. In the oceanic water-mass decomposition problem (e.g., Broecker and Peacock, 1998), the number of water masses is often chosen to be equal to the number of observations plus one, so that the inversion appears just determined. (One additional water mass can be determined due to the constraint of conservation of mass.) A long literature exists for decomposing the Pacific Ocean into two water masses: one of Antarctic and one of North Atlantic origin. The seemingly well-posed problem gives an equipartition of the deep ocean into the 2 water masses. When one more observation is added, the abundance of one more water mass can be solved, and the amount of North Atlantic water is halved. As more observations are added, the amount of North Atlantic water converges to about 25%, much less than the 2-water-mass case. Thus, we suggest that many oceanographers have been misled by overly restricting the number of parameters to be estimated. (Munk's (1966) Abyssal Recipes paper is another example.) Furthermore, the least-squares error estimates do not reflect this error, because it is a systematic one not captured in the model equations. Thus, our philosophy is that the restriction of the number of controls is potentially dangerous to scientific interpretation.

The examples presented above are rather specific, and other real-world problems may behave quite differently, especially when the error statistics are not accurately known. That being said, we don't know what examples the reviewer has in mind, and thus we can't evaluate the statement that "the dimension of the control space should not exceed the number of observations." We do know a number of examples shown above that contradict this statement.

- *As I said before, it is good to see the chi-square test used systematically in this way, but the authors should note that it is J (equation 6) rather than $J_d$ (equation 4) that is the useful chi-square random variable, the randomness arising only from the observation noise. If only $J_d$ is considered, there will be no indication if the prior control estimates and their statistics were reasonable; see the texts by Bennett or Kalnay.*

We thank the reviewer for catching this subtle but important detail! To improve the manuscript, we have re-created Figure 6 for $J$ instead of $J_d$. We have also run the exploration of phase space over a wider range. The basic story of the Figure remains the same, but some of patterns are smoother than the original.

- *p5, line 1: Sx and Sf are weights that restrict the size of the perturbations to 5rad and 10rad s2, respectively. I wouldnt say restrict. Greater perturbations are permitted, but penalized more severely.*

To take the reviewer's suggestion into account, we've revised the description to simply say that deviations are penalized.

> A second contribution to the cost function includes two terms that constrain the difference between our posterior and prior estimates of the initial conditions and forcing,

$$J_0 = [\mathbf{x}(0) - \mathbf{x}_0(0)]^T \mathbf{S}_x^{-1} [\mathbf{x}(0) - \mathbf{x}_0(0)] + \sum_{i=0}^{N_t-1} [f(i\Delta t) - f_0(i\Delta t)]^T S_f^{-1} [f(i\Delta t) - f_0(i\Delta t)] \quad (1)$$

> where $\mathbf{x}_0(0)$ is the first-guess initial conditions, there are $N_t$ model timesteps, $f_0(t)$ is the first-guess forcing, and $\mathbf{S}_x$ and $S_f$ are weights of 5 rad and 10 rad s$^{-2}$, respectively, that penalize deviations.

- *p6, line 20: Please give the definition of the controllability matrix, and state the theorem that specifies the condition for controllability. If I dont remember, neither do 90% of the readers of NPG. Its not reasonable to send your readers scurrying to the literature to find those things at this point.*

Equation (8) gives the full definition of the controllability matrix. Equation (9) then rewrites equation (8) in a symbolic form and defines the controllability matrix. It is explicitly stated in the line that references Dahleh and Diaz-Bobillo.

- *p7 The method of total inversion needs some further explanation. The terminology is not standard in the community that constitutes the readership of NPG. Without further explanation, its hard to know what to make of (12). Consider the expression in the first square brackets. The matrix E is 1x2 (p4, line 24) and the matrix Q appears to be 3x3. Its not even clear that (12) is a valid vector expression. Also, "we update the state transition and controllability matrices iteratively." Since that is the case, you probably want the last expression to reflect the evolution state from $i\Delta t_y$ to $(i+1)\Delta t_y$ (pardon my TeX, but I dont see how to put symbols into this writeup)*

Thanks to the reviewer for catching the error in equation (12). The proper equation (12) is:

$$\mathbf{u} = \mathbf{Q}\mathbf{C}^T\mathbf{E}^T[\mathbf{E}\mathbf{C}\mathbf{Q}\mathbf{C}^T\mathbf{E}^T + W]^{-1}\{y(\Delta t_y) - \mathbf{E}\mathcal{R}(\Delta t_y, 0)[\mathbf{x}_0(0)]\}. \qquad (2)$$

Now the terms in Equation (12) have consistent dimensions. $\mathbf{E}$ is $1 \times 2$, $\mathbf{C}$ is $2 \times (K+2)$, $\mathbf{Q}$ is $(K+2) \times (K+2)$, $\mathbf{W}$ is $1 \times 1$, and $\mathbf{u}$ is $(K+2) \times 1$. Yes, the reviewer is correct about the timestepping from $i\Delta t$ to $(i+1)\Delta t$, but the notation becomes cluttered if we make the equation more general in this way. Instead, we just show the equation for the first segment, and then state in words that it must be repeated for the other segments.

To make the writing more logical, clear, and accessible, we have rewritten this whole section as follows.

In the case of an underdetermined problem, we solve for $\mathbf{u}$ with the least-squares formula,

$$\mathbf{u} = \mathbf{Q}\mathbf{C}^T\mathbf{E}^T[\mathbf{E}\mathbf{C}\mathbf{Q}\mathbf{C}^T\mathbf{E}^T + W]^{-1}\{y(\Delta t_y) - \mathbf{E}\mathcal{R}(\Delta t_y, 0)[\mathbf{x}_0(0)]\}, \qquad (3)$$

but note the nonlinearity due to $\mathcal{R}(\Delta t_y, 0)$. To handle this quantity, we update the state transition and controllability matrices iteratively, which is identical to the method of total inversion (*Tarantola and Valette*, 1982). In cases where it saves computations, we employ the overdetermined least-squares formula instead of (3). The full nonlinear model is run with the updated controls to produce the improved first-guess trajectory, $\mathbf{x}_1(t)$, for the first segment ($0 \leq t \leq \Delta t_y$). The algorithm proceeds sequentially $N_y - 1$ times, where the terminal state from one segment becomes the initial condition for the next.

- *Starting with the improved first guess and then iterating until a satisfactory chi-square statistic is achieved is a good strategy. I have seen very, very few inversions of realistic models of the ocean or atmosphere that satisfy the chi-square condition. Its really the chi-square statistics that allow us to conclude that the black curve in the bottom panel of figure 3 is to be preferred to the gray curve. The real world is rarely so accommodating. The last sentence of section 3.2, The method of Lagrange multipliers is therefore superior to the first-guess estimate because all components of the solution, both the state and forcing, can be physically interpreted. is only meaningful to the extent that the chi-square statistics say it is.*

Thanks for reminding us of this salient point. We think our discussion is improved by adding the final sentence of the following quote.

The suppression of the abrupt and large changes in forcing is not simply a smoothing or averaging of the forcing, but instead is seen to reflect the true forcing, as evidenced by the deviation from true forcing being small. Strictly speaking, a $\chi^2$ posterior statistical test, discussed later in Section 3.4, is needed to assess what is meant by "small."

- *p9: small-scale features that are added to the surface forcing of ocean models in order to fit observations (e.g., Stammer et al., 2002b) The real problem with the reconstructed forcing in Stammer et al. (2002b) was not small scale features but large scale features because of inaccurate model dynamics.*

Thanks to the reviewer for the interesting observation. Should the opportunity arise, we'd like to learn about the evidence that supports the reviewer's claim, as we are unaware of anything in the published literature. Alas, the structure of peer review might make such information exchange difficult!

- *p10: As is strongly implied in the following discussion, the statement that there appears to be no lower limit on the number of observations necessary in order to produce an acceptable state estimate with this model is only valid if there are sufficient degrees of freedom in the control space. The Lagrange multiplier method would not be nearly so successful if the control space were restricted to the initial condition, as was the case in some of the earlier studies that cast doubt on the efficacy of 4DVAR in chaotic systems. This point is nicely made in figures 2 and 7 of the present manuscript.*

Indeed, the reviewer is correct and has nicely summarized some of our key points.

- *p10: the null hypothesis that the model is consistent with observations. It is true that, for the area above the heavy line in figure 6 a satisfactory fit to data can be obtained, but a too-low chi square statistic is evidence of overfitting, and the reconstructed inputs cannot be trusted.*

  Thanks for this nice suggestion. We now include the following paragraph in Section 3.4.

    For some cases where only 5 or 10 observations are available, the cost function is small enough that overfitting may be occurring. In these cases, we find that the control perturbations necessary to fit the data are very small, and this impacts the size of the cost function through the $J_0$ term in equation (6). This effect has been documented in chaotic systems by the control engineering literature (e.g., *Ott et al.*, 1990).

- *Also, figure 6 depicts a contour plot of $J_d$. It does not, therefore, distinguish among different choices of forcing fields.*

  Correct, this figure solely focused on $J_d$ in the original manuscript. As mentioned above, we have now remade it with $J$ so that the $\chi^2$ test is more rigorous.

- *last sentence: In the ocean state estimation problem, all air-sea fluxes are uncertain and temporally variable so there is no shortage of controls that can be defined. That is true, but amount of independent air-sea flux data is limited. There may be no shortage of controls that can be defined, but there is a very real shortage of controls that can be constrained by observation.*

  This is a good thing to keep in mind, so we've added it to the body of the text.

    Alternatively, this could be accomplished using a nondiagonal weighting matrix, $\mathbf{S}_f$. In this case, the number of degrees of freedom is reduced relative to the total number of controls.

  We also temper our statement in Section 3.4.

    In the ocean state estimation problem, all air-sea fluxes are uncertain and temporally variable, so a large number of controls can usually be defined.

- *p12: Relation to the Kalman filter/smoother. Line 21: Both methods solve the same least squares problem This is not strictly true. The Kalman smoother solves the same least squares problem as 4DVAR, but the Kalman filter does not. The Kalman filter and 4DVAR are only guaranteed to agree at the final time.*

  We are referring to the combined Kalman filter/smoother, as discussed by Wunsch (2006). It does solve the same least squares problem as the method of Lagrange multipliers in the linear case. We now emphasize that we are talking about the combined filter/smoother by adding the following to the text:

    ... into closer consistency with the Kalman filter/smoother (i.e., the combined usage of the Kalman filter and smoother)

- *Figure 9 gives a nice illustration of the effect of nonlinearity.*

  Thank you for the encouraging comment regarding Figure 9.

[revised manuscript text omitted]